# Planthopper salivary sheath protein LsSP1 contributes to manipulation of rice plant defenses

Hai-Jian Huang [1,2], Yi-Zhe Wang[1,2], Li-Li Li[1,2], Hai-Bin Lu[1,2], Jia-Bao Lu[1,2], Xin Wang[1,2], Zhuang-Xin Ye[1,2], Ze-Long Zhang[1,2], Yu-Juan He[1,2], Gang Lu[1,2], Ji-Chong Zhuo[1,2], Qian-Zhuo Mao[1,2], Zong-Tao Sun[1,2], Jian-Ping Chen [1,2] ✉, Jun-Min Li [1,2] ✉ & Chuan-Xi Zhang [1,2] ✉

Salivary elicitors secreted by herbivorous insects can be perceived by host plants to trigger plant immunity. However, how insects secrete other salivary components to subsequently attenuate the elicitor-induced plant immunity remains poorly understood. Here, we study the small brown planthopper, *Laodelphax striatellus* salivary sheath protein LsSP1. Using Y2H, BiFC and LUC assays, we show that LsSP1 is secreted into host plants and binds to salivary sheath via mucin-like protein (LsMLP). Rice plants pre-infested with ds*LsSP1*-treated *L. striatellus* are less attractive to *L. striatellus* nymphs than those pre-infected with ds*GFP*-treated controls. Transgenic rice plants with LsSP1 over-expression rescue the insect feeding defects caused by a deficiency of LsSP1 secretion, consistent with the potential role of LsSP1 in manipulating plant defenses. Our results illustrate the importance of salivary sheath proteins in mediating the interactions between plants and herbivorous insects.

In nature, plants are continuously challenged by various pathogens, including bacteria, fungi, and nematodes. To survive or fend off attacks, plants have evolved multi-layered immune systems from recognizing pathogens to activating defense responses. Pattern recognition receptors can perceive "non-self" molecules and activate the pattern-triggered immunity (PTI), including mitogen-activated protein kinase (MAPK) cascades, reactive oxygen species (ROS), and hormone signaling[1–3]. To counteract plant immunity, plant pathogens deliver secretory effectors to target the immune signaling components of PTI and interfere with their activities[4,5]. However, some effectors can be sensed by the plants with time, further initiating the effector-triggered immunity[6]. Over millions of years of co-evolution, plant pathogens have developed dynamic and complex interactions with host plants.

Piercing-sucking insects, such as planthoppers, aphids, and whiteflies, are important pests that damage host plants by feeding or transmitting viruses. During the feeding process, two types of saliva (gel saliva and watery saliva) are ejected into plant tissues[7]. These oral secretions, on the one hand, hinder insect performance by activating plant defenses. For example, salivary protein Cathepsin B3 from *Myzus persicae* can be recognized by *Nicotiana tabacum* plants, which thus suppress aphid feeding by triggering ROS accumulation[8]. Moreover, salivary protein 1 from *Nilaparvata lugens* induces cell death, $H_2O_2$ accumulation, defense-related gene expression, and callose deposition when it is transiently expressed in *Nicotiana benthamiana* leaves or rice protoplasts[9]. On the other hand, saliva exerts multiple roles in improving insect performance, such as calcium binding proteins for calcium regulation[10], DNase II for extracellular DNA degradation[11], and *Helicoverpa armigera* R-like protein 1 (HARP1) for plant hormonal manipulation[12]. Piercing-sucking herbivores eject abundant salivary effectors into plant tissues. There may be some salivary elicitors triggering plant defenses, while the elicitor-induced plant defenses are inhibited by other salivary components. However, little is known about the complex interactions in saliva.

[1]State Key Laboratory for Managing Biotic and Chemical Threats to the Quality and Safety of Agro-products, Institute of Plant Virology, Ningbo University, 315211 Ningbo, China. [2]Key Laboratory of Biotechnology in Plant Protection of MARA and Zhejiang Province, Institute of Plant Virology, Ningbo University, 315211 Ningbo, China. ✉e-mail: jpchen2001@126.com; lijunmin@nbu.edu.cn; chxzhang@zju.edu.cn

Formed by gel saliva, salivary sheath is indispensable for insect feeding. It is secreted during the stylet probing, which provides mechanical stability and lubrication for stylet movement[13]. The salivary sheath is capable of sealing the stylet penetration site, thereby preventing the plant immunity triggered by leaked cell components[7]. In aphids and planthoppers, the disrupted salivary sheath formation can hinder insect feeding from plant sieve tubes, but not from the artificial diet[14,15]. After secretion, salivary sheath is distributed in plant apoplast and directly contacts with plant cells[15]. Salivary sheath is composed by many salivary sheath proteins, which can be potentially recognized as herbivore-associated molecular patterns (HAMPs) that activate the immune response in host plants[16]. Because of the forward roles in herbivore–plant interactions, a few proteins in the salivary sheath may exhibit a high evolutionary rate[17]. Nevertheless, the current knowledge on salivary sheath is mainly limited on its mechanical function. Therefore, it is interesting to reveal its other functions in herbivore–plant interactions.

Plant apoplast space is an important battleground between the host and pathogens[18]. The papain-like cysteine proteases (PLCPs), which share a conserved protease domain, are prominent enzymes in the plant apoplast that can function as the central hubs in plant immunity[19]. As a well-known maize insect resistance gene, Mir1 belongs to PLCPs[20]. It can be rapidly accumulated at the wound sites, and can degrade the insect gut surface to confer maize resistance against caterpillars[20,21]. Mir1 accumulation is reported to enhance plant resistance against root-feeding herbivores and corn leaf aphids[22,23]. In turn, PLCPs are the common targets of pathogen effectors. Fungi, oomycete, nematodes, and bacteria can actively interfere with the activity or subcellular location of plant PLCPs, which can thereby suppress plant immunity[24–28].

The small brown planthopper, *Laodelphax striatellus*, is a destructive pest that causes severe yield reductions and economic losses in rice crops. Similar to most phloem-feeding insects, planthoppers can secrete a mixture of saliva during feeding. Several salivary proteins have been found to participate in salivary sheath formation and/or interfere with the host immune responses[13,29]. Nevertheless, the functions of most salivary proteins remain unknown. In this study, *L. striatellus* salivary sheath protein LsSP1 is employed as a molecular probe to investigate the mechanism by which this planthopper can interact with salivary sheath mucin-like protein (LsMLP)-triggered, PLCP-mediated plant defenses. The salivary LsSP1 is secreted into host plants during feeding and is shown to interact with multiple PLCPs belonging to different subfamilies in yeast two hybrid (Y2H) and bimolecular fluorescence complementation (BiFC) assays. OsOryzain is a member of PLCPs. Expression of LsSP1 in *N. benthamiana* plants significantly attenuates the $H_2O_2$ accumulation and defense gene expression induced by OsOryzain and LsMLP, while in rice plants the role of OsOryzain is not confirmed. Overexpression of LsSP1 in rice plants rescues the feeding defects caused by a deficiency in LsSP1 secretion.

## Results

### LsSP1 is important for *L. striatellus* feeding on rice plants

Many genes that highly expressed in *L. striatellus* salivary glands were planthopper-specific[30], and their homologous genes were not found in other species (Supplementary Data 1). To reveal their specific roles in the planthopper-rice interactions, this study firstly investigated the expression patterns of these genes in different tissues. In total, 30 genes were found to be specifically expressed in salivary glands (Supplementary Fig. 1). The *L. striatellus* salivary protein 1 (hereafter: LsSP1, accession number: ON322955) was among the top 5 most abundant, salivary gland-specific, and planthopper-specific genes, which was therefore selected for further analysis. The insect survivorship was not significantly affected by treating *L. striatellus* with ds*LsSP1* (log-rank test, $p = 0.3044$; Fig. 1a). However, the ds*LsSP1*-treated *L. striatellus* produced less offspring (one-way ANOVA test followed by Tukey's multiple comparisons test, $p = 0.0153$; Fig. 1b) and excreted less honeydew (one-way ANOVA test followed by Tukey's multiple comparisons test, $p = 0.0127$; Fig. 1c) than the ds*GFP*-treated control. Electrical penetration graph (EPG) was adopted for monitoring the insect feeding behavior. Compared with ds*GFP* treatment, *L. striatellus* treated with ds*LsSP1* exhibited a significant decrease (by 62%; two-tailed unpaired Student's $t$ test, $p = 0.0057$) in phloem sap ingestion, along with the slight increases in nonpenetration (by 23%; two-tailed unpaired Student's $t$ test, $p = 0.2769$) and pathway duration phase (by 31%; two-tailed unpaired Student's $t$ test, $p = 0.2525$) (Fig. 1d, e). These results indicate that LsSP1 plays a role in *L. striatellus* feeding on rice plants.

### LsSP1 is a salivary sheath protein not essential for salivary sheath formation

*LsSP1* contained an open reading frame of 771 bp, encoding a protein of 256 amino acids. No conserved domain was found in LsSP1. The protein possessed an N-terminal signal peptide, with no transmembrane domain, which indicated its secretory property (Supplementary Fig. 2a). Homologous analysis demonstrated that LsSP1 was a planthopper-specific protein, and exhibited 43.9% and 58.5% amino acid sequence identities to secretory proteins in the brown planthopper *N. lugens* (ASL05017) and the white-backed planthopper *Sogatella furcifera* (ON322954), respectively (Supplementary Fig. 2b). LsSP1 and its homologous genes in other planthopper species have not been well investigated previously. Spatial-temporal expression analysis showed that LsSP1 was mainly expressed at the nymph and adult stages (Supplementary Fig. 2c), and immunohistochemical (IHC) staining showed that LsSP1 was exclusively expressed in a pair of follicles in primary salivary glands (Fig. 2a, b). The transcript level of *LsSP1* was reduced by 90% after the treatment of *L. striatellus* with ds*LsSP1*, and almost no LsSP1 signal was detected in salivary glands (Supplementary Fig. 3a–c). LsSP1 was secreted during insect feeding, and a band of approximately 35 kDa was detected in rice plants infested by *L. striatellus*, but not in the non-infested plants (Fig. 2c).

For most of the piercing-sucking insects, two types of saliva (gel and watery saliva) are ejected into plant tissues during the feeding process. Previously, the components of *L. striatellus* watery saliva collected by artificial diet were reported[30]. However, LsSP1 was not detected in those samples. Thereafter, the salivary sheath (gel saliva) was collected from the inner layer of the Parafilm membrane to investigate whether LsSP1 existed in the salivary sheath. As a result, a band of LsSP1 was detected in the salivary sheath sample (Fig. 2c). By contrast, the band of LsSP1 in watery saliva sample was not visible, indicating that LsSP1 was a salivary sheath protein. Immunohistochemistry (IHC) staining analysis of salivary sheath on the Parafilm membrane and in rice plants further confirmed the presence of LsSP1 in salivary sheath (Fig. 2d, e), whereas almost no signal was detected in salivary sheath secreted from ds*LsSP1*-treated *L. striatellus* (Supplementary Fig. 3d, e). LsSP1 deficiency did not influence salivary sheath formation, and there was no significant difference in salivary sheath appearance between ds*LsSP1* treatment and the control as observed under scanning electron microscopy (SEM; Supplementary Fig. 4). Also, we did not find significant difference in the length of salivary sheath on the Parafilm membrane (two-tailed unpaired Student's $t$ test, $p = 0.5926$; measured from the top to base of salivary sheath under SEM) or the number of salivary sheaths left on the rice surface (two-tailed unpaired Student's $t$ test, $p = 0.7615$; measured by counting the ring-shaped salivary sheath structure under SEM) after ds*LsSP1* treatment (Supplementary Fig. 4). These results suggest that LsSP1 is a salivary sheath protein, but that it is not indispensable for salivary sheath formation, which is significantly different from two previously reported salivary sheath proteins[14,31].

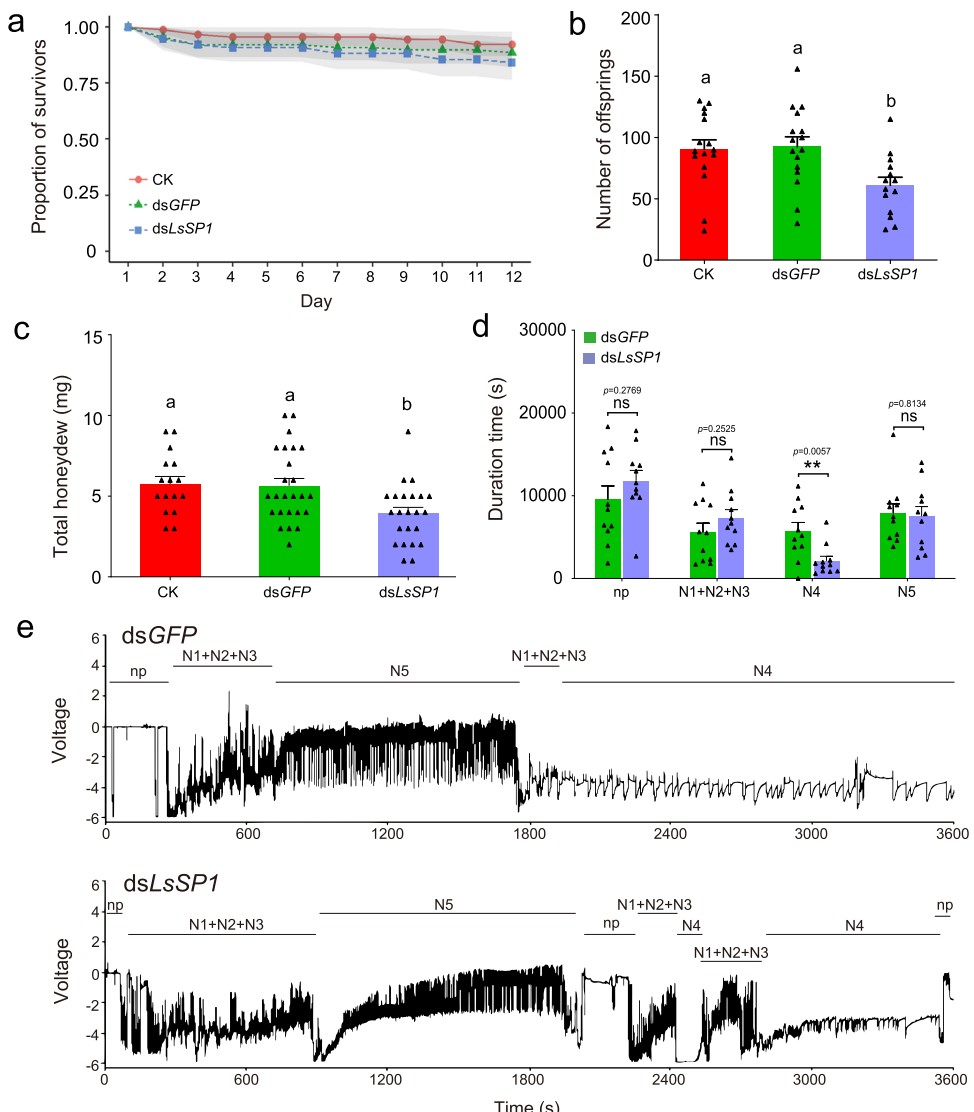

**Fig. 1 | Effect of dsRNA treatment on *Laodelphax striatellus*.** **a**–**c** Effect of *LsSP1* knockdown on *L. striatellus* survival rate (**a**), fecundity (**b**), and honeydew excretion (**c**). The untreated (CK) and ds*GFP*-treated *L. striatellus* were used as controls. Data in **a** are presented as mean values ±95% confidence intervals (displayed in light shades). Different lowercase letters indicate statistically significant differences at $P < 0.05$ level according to log-rank test (**a**) or one-way ANOVA test followed by Tukey's multiple comparisons test (**b**, **c**). **d** Comparison of electrical penetration graph (EPG) parameters between ds*GFP*-treated and ds*LsSP1*-treated *L. striatellus*. All EPG recordings were performed for 8 h. *P*-values were determined by two-tailed unpaired Student's *t* test. **P < 0.01; ns, not significant. Data in **b**–**d** are presented as mean values ±SEM. For survival analysis in **a**, $n = 91$, $n = 88$, and $n = 106$ individuals in CK, ds*GFP*, and ds*LsSP1*, respectively; for fecundity analysis in **b**, $n = 20$ independent biological replicates in three treatments; for honeydew analysis in **c**, $n = 16$, $n = 25$, and $n = 23$ independent biological replicates in CK, ds*GFP*, and ds*LsSP1*, respectively; for EPG analysis in **d**, $n = 11$ independent biological replicates in three treatments. **e** Overall typical EPG waveforms over 1 h for ds*GFP*-treated (upper) and ds*LsSP1*-treated *L. striatellus* (lower). The insect feeding behavior was classified into nonpenetration (np), pathway duration (N1 + N2 + N3), phloem sap ingestion (N4), and xylem sap ingestion (N5) phases. The rice variety cv. ASD7 was used. Source data are provided as a Source Data file.

## LsSP1 binds to the salivary sheath protein mucin-like protein LsMLP using Y2H, BiFC and LUC assays

Our previous work demonstrated that mucin-like protein (MLP) was the main component of salivary sheath in the planthopper *N. lugens*[31]. Amino acid alignment demonstrated that MLPs among three planthoppers were highly homologous (Supplementary Fig. 5a). At first, the function of *L. striatellus* MLP (LsMLP, accession number: ON568348) was investigated by RNAi (Supplementary Fig. 5b). The LsMLP-deficient *L. striatellus* only secreted the short salivary sheath (two-tailed unpaired Student's *t* test, $p < 0.001$; Supplementary Fig. 6), similar to that of NlMLP-deficient *N. lugens*[16]. The number of salivary sheaths left on the rice plant significantly decreased when *L. striatellus* was treated with ds*LsMLP* (two-tailed unpaired Student's *t* test, $p = 0.015$; Supplementary

Fig. 6). Furthermore, the LsMLP-deficient *L. striatellus* exhibited a high mortality rate (log-rank test, $p < 0.001$; Supplementary Fig. 5c), indicating that LsMLP is important for *L. striatellus* performance.

Meanwhile, the treatment of *L. striatellus* with ds*LsMLP* did not influence LsSP1 at the transcript (two-tailed unpaired Student's *t* test, $p = 0.5317$; Fig. 3a) or the protein level (Fig. 3b). However, almost no fluorescence signal of LsSP1 was detected in salivary sheath secreted from ds*LsMLP*-treated *L. striatellus*, which was significantly different from that of ds*GFP*-treated control (Fig. 3c). Thereafter, this study detected whether LsSP1 existed in watery saliva or salivary sheath secreted from ds*LsMLP*-treated *L. striatellus*. Interestingly, more LsSP1 was found in watery saliva than in salivary sheath collected from ds*LsMLP*-treated *L. striatellus*,

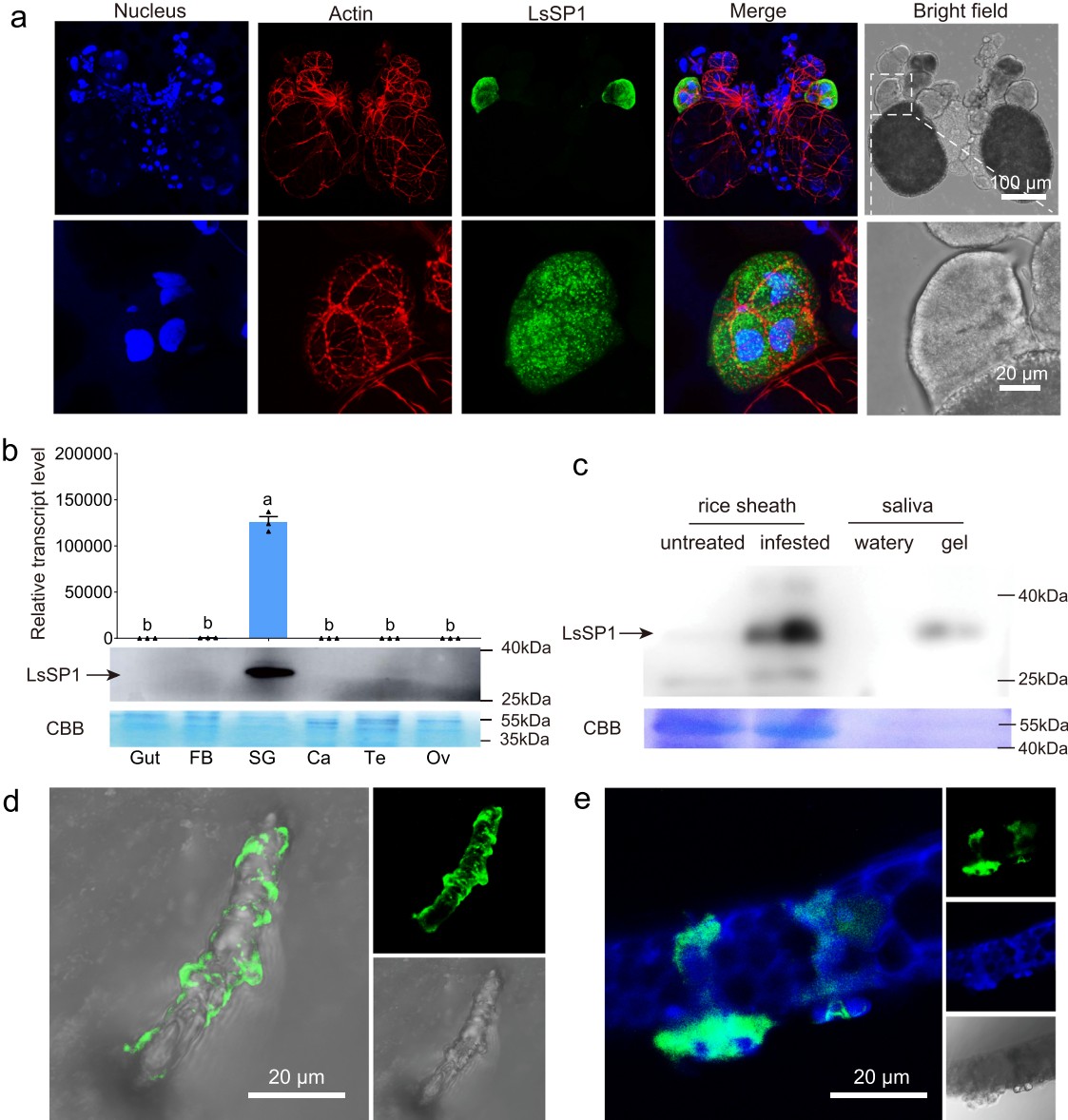

**Fig. 2 | LsSP1 is a salivary sheath protein and secreted into plants.**
**a** Immunohistochemical staining of LsSP1 in salivary glands. *L. striatellus* salivary glands were incubated with anti-LsSP1 serum conjugated with Alexa Fluor™ 488 NHS Ester (green) and actin dye phalloidinrhodamine (red) and examined by Leica SP8. The nucleus was stained with DAPI (blue). The lower images represent the enlarged images of the boxed area in the upper image. The boxed area was indicated in bright filed image. **b** Expression patterns of LsSP1 in different tissues quantified by qRT-PCR (upper) and western-blotting (lower) assays. FB fat body, SG salivary gland, Ca carcass, Te testis, Ov ovary. Data are presented as mean values ±

SEM (*n* = 3 independent biological replicates). Different lowercase letters indicate statistically significant differences at *P* < 0.05 level according to one-way ANOVA test followed by Tukey's multiple comparisons test. **c** Detection of LsSP1 in untreated plants (lane 1) and plants infested by *L. striatellus* (lane 2), watery (lane 3) and gel saliva (lane 4). **d, e** LsSP1 staining of salivary sheath on parafilm (**d**) and in rice tissues (**e**). Green, LsSP1; blue, nucleus. Coomassie brilliant blue (CBB) staining was conducted to visualize the amount of sample loading. Experiments in **a**, **c**, **d**, and **e** were repeated three times with the similar results. Source data are provided as a Source Data file.

perhaps indicating that LsSP1 fails to bind to salivary sheath in the absence of LsMLP (Fig. 3d).

The potential interaction between LsSP1 and LsMLP was investigated using point-to-point Y2H assays. The yeast transformants expressing DNA-binding domain (BD)-LsMLP and activating domain (AD)-LsSP1 were found to grow on the quadruple dropout medium, which was not observed in transformants bearing the control constructs (Fig. 3e). Similar results were found in yeast transformants expressing BD-LsSP1 and AD-LsMLP (Fig. 3e). Also, the interaction between LsSP1 and LsMLP was verified by BiFC assay (Fig. 3f) and luciferase complementation (LUC) assay (Fig. 3g, h). These results may suggest that LsSP1 interacts with LsMLP in vivo.

## LsSP1 can interact with rice papain-like cysteine proteases using Y2H, GST-pull down, BiFC, and LUC assays

To understand the potential roles of LsSP1 in insect-plant interaction, Y2H screening was performed using a rice cDNA library. Seven proteins were found to potentially interact with LsSP1, including an *Oryza sativa* Oryzain (OsOryzain, NP_001389372.1, LOC_Os04g55650) (Supplementary Table 1). OsOryzain was highly homologous with Arabidopsis RD21, tomato C14, and maize Mir3 cysteine proteases. It contained a predicted N-terminal secretion signal and a self-inhibitory prodomain followed by the peptidase and granulin domains (Supplementary Fig. 7a). OsOryzain is a member of PLCPs that act as a central hub in plant immunity and are required for the full resistance of plants

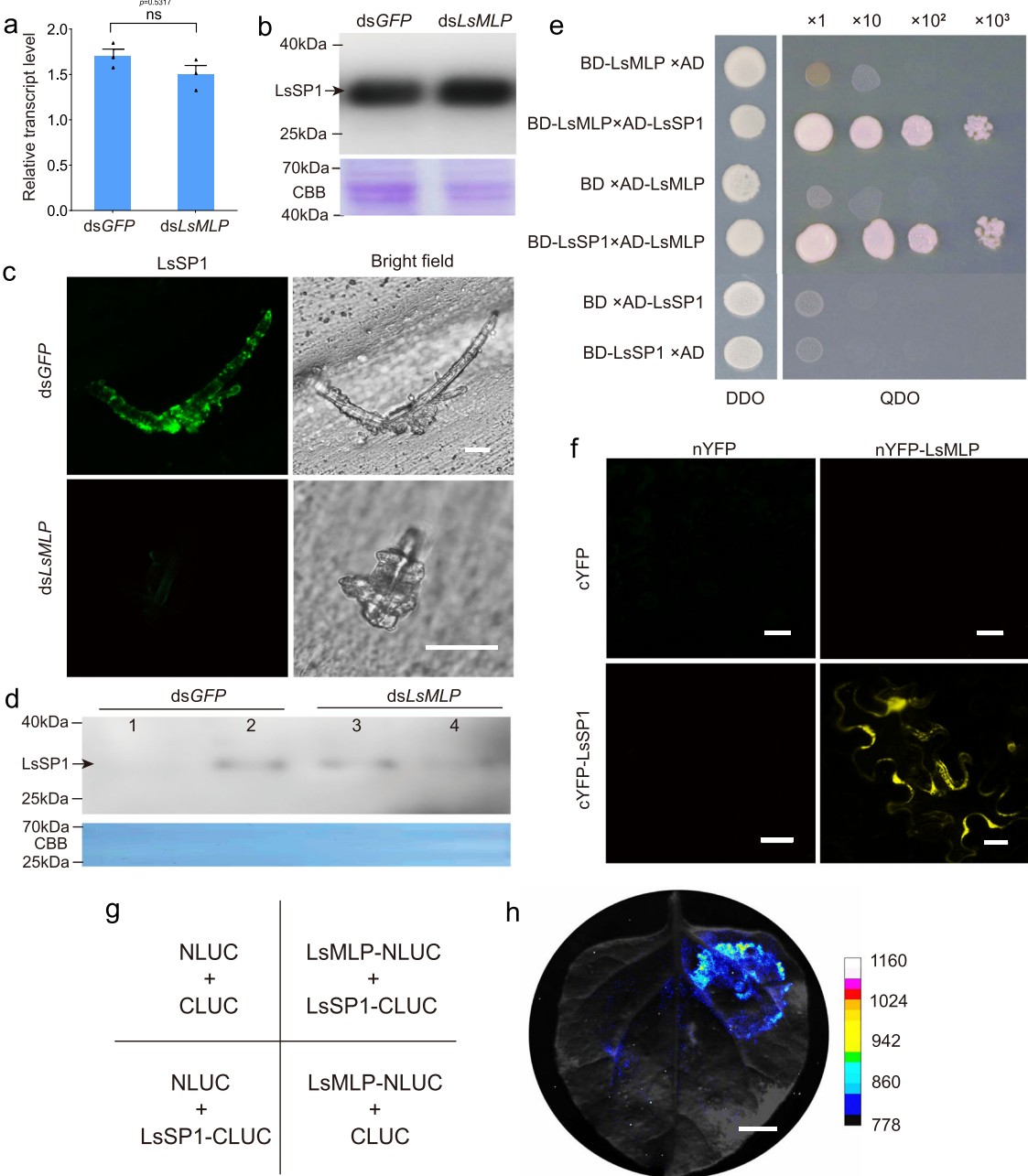

**Fig. 3 | LsSP1 binds to a salivary sheath protein LsMLP. a, b** Influence of ds*LsMLP* treatment on the LsSP1 abundance in salivary gland quantified by qRT-PCR (**a**) and western-blotting (**b**) assays. Data are presented as mean values ±SEM (*n* = 3 independent biological replicates). *P*-values were determined by two-tailed unpaired Student's *t* test. ns, not significant. **c** LsSP1 staining of salivary sheath secreted from dsRNA-treated *Laodelphax striatellus*. The salivary sheaths secreted from ds*GFP*- or ds*LsMLP*-treated *L. striatellus* were probed with the anti-LsSP1 serum conjugated with Alexa Fluor™ 488 NHS Ester (green), and examined by Leica SP8. Bars = 20 μm. **d** Detection of LsSP1 in watery (lane1, lane3) and gel saliva (lane 2, lane 4) secreted from ds*GFP*- or ds*LsMLP*-treated *L. striatellus*. **e** Yeast two hybrid assays showing the interaction between LsSP1 and LsMLP. The different combinations of constructs transformed into yeast cells were grown on the selective medium SD/-Trp/-Leu (DDO), and the interactions were tested with SD/-Trp/-Leu/-His/-Ade (QDO). **f** Bimolecular fluorescence complementation (BiFC) assays showing the interaction between LsSP1 and LsMLP. Bars = 20 μm. **g** The co-expression scheme in *Nicotiana benthamiana* leaves during Luciferase complementation (LUC) assays. **h** Results from LUC assays showing the interaction between LsSP1 and LsMLP. Bar = 1 cm. Experiments in **b**, **c**, **d**, **f**, and **h** were repeated three times with the similar results. Source data are provided as a Source Data file.

to various pathogens[19]. In tomato, C14 is converted into immature (iC14) and mature (mC14) isoforms that are accumulated into various subcellular compartments and the apoplast[28]. The interaction between OsOryzain and LsSP1 was confirmed using point-to-point Y2H, GST-pull down, BiFC, and LUC assays (Supplementary Note 1 and Supplementary Fig. 7b–f). Furthermore, our experiments show that LsSP1 is capable of interacting with multiple PLCPs belonging to different subfamilies using point-to-point Y2H and BiFC assays (Supplementary Note 1 and Supplementary Figs. 8 and 9).

## Induction of PLCPs by *L. striatellus* infestation and salicylic acid (SA) treatment using rice plants
Relative transcript levels of PLCPs in response to *L. striatellus* infestation were investigated. Among the 46 PLCPs investigated, 8 were

found to be significantly induced upon *L. striatellus* infestation (Supplementary Fig. 10). The expression of *OsOryzain* was induced at 3 h post-infestation, and reached a peak at 6 h (Supplementary Fig. 11a). Salicylic acid (SA) exerts a critical role in plant defense against sap-sucking herbivores[32–34]. The induction of SA biosynthetic genes and SA responsive genes was detected upon *L. striatellus* infestation (Supplementary Fig. 12). To investigate the possible role of SA in regulating PLCPs, the relative transcript levels of PLCPs were quantified after SA treatment. As a result, SA significantly induced the expression of 7 PLCPs, including OsOryzain (Supplementary Figs. 10 and 11a). These results indicate that numerous PLCPs might be associated with the *L. striatellus*-induced SA-mediated plant defenses in rice plants.

In addition, our experiments also investigated the protein levels of OsOryzain in response to SA treatment and *L. striatellus* infestation. The results demonstrated that SA treatment and *L. striatellus* infestation induced the expression of OsOryzain in plant cells, while rice plants infested by *L. striatellus* secreted a lower amount of mature OsOryzain (mOsOryzain) into apoplast than that under SA treatment (Supplementary Note 2 and Supplementary Fig. 11b, c). We were not able to confirm that OsOryzain is involved in the plant defense response to *L. striatellus*, hence additional methods and results

involving OsOryzain experiments have been moved to the Supplementary Note 3–4, Supplementary Methods, and Supplementary Figs. 13–15.

## LsSP1 affects plant defenses in rice plants

To determine whether LsSP1 affects plant defenses in rice plants, the feeding preference of *L. striatellus* nymphs on plants pre-infested by ds*GFP*- and ds*LsSP1*-treated *L. striatellus* was compared. The results revealed that rice plants pre-infested with ds*LsSP1*-treated *L. striatellus* were less attractive to *L. striatellus* nymphs than those pre-infested with ds*GFP*-treated controls (Fig. 4a), suggesting that ds*LsSP1*-treated *L. striatellus* might elicit plant defenses and become less palatable to conspecifics.

Thereafter, plants infested by ds*GFP*-treated *L. striatellus* and ds*LsSP1*-treated *L. striatellus* were subject to transcriptomic sequencing. In total, 405 differentially expressed genes (DEGs) were identified, among which 90.9% were up-regulated in ds*LsSP1*-treated *L. striatellus* infested plants (Supplementary Fig. 16 and Supplementary Data 2). Enrichment analysis demonstrated that the majority of DEGs were involved in plant-pathogen interaction, environmental adaptation, transporters, plant hormone signal transduction, and terpenoids metabolism (Fig. 4b). Among the 28 SA biosynthetic or SA responsive

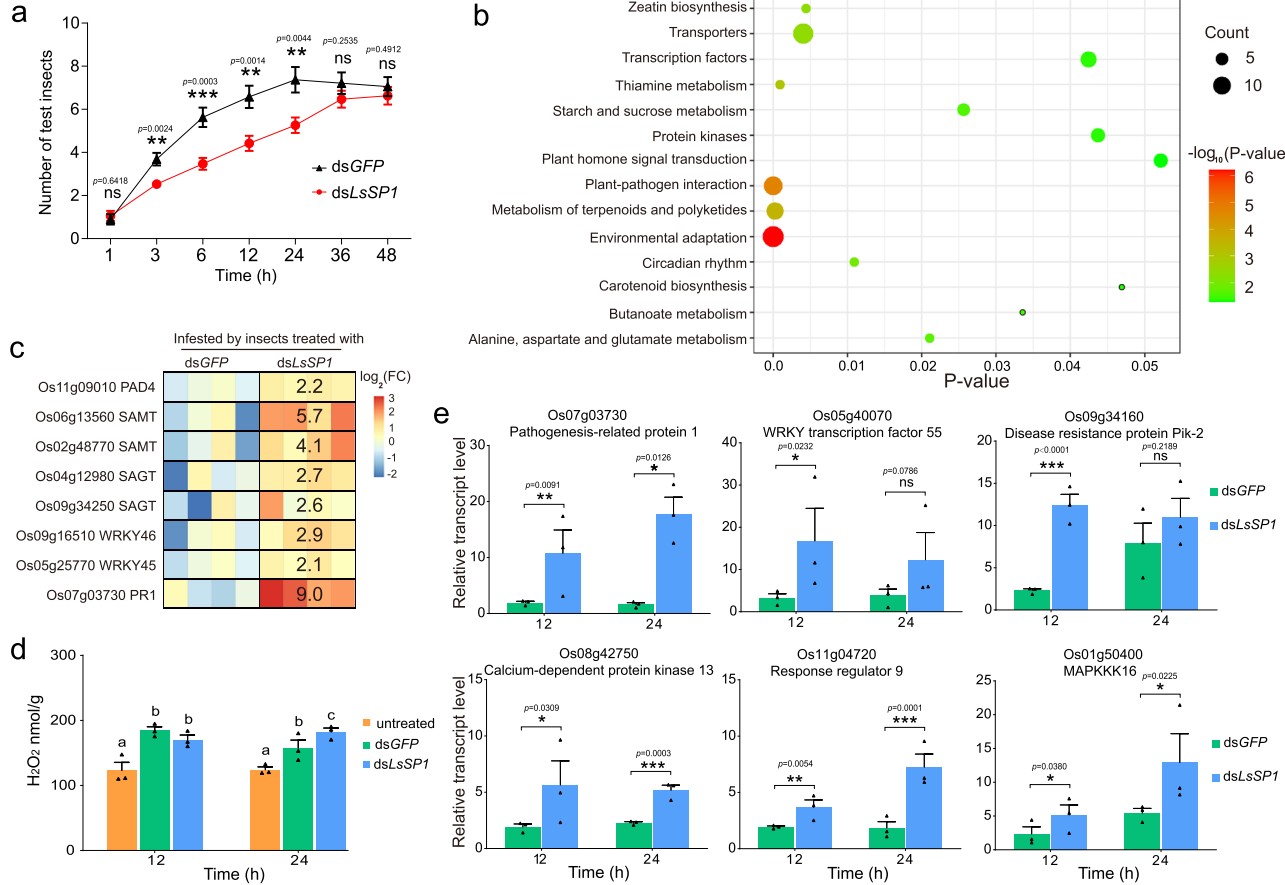

**Fig. 4 | Influences of dsRNA-treated *Laodelphax striatellus* on rice plants. a** The attraction of rice plants infested by dsRNA-treated *L. striatellus* to nymphs in the two-choice equipment. Data are presented as mean values ±SEM (*n* = 19 independent biological replicates). *P*-values were determined by two-tailed unpaired Student's *t* test. ***P* < 0.01; ****P* < 0.001; ns, not significant. **b** Kyoto Encyclopedia of Genes and Genomes (KEGG) pathway enrichment analysis of differentially expressed genes (DEGs). Enriched *P*-values were calculated according to one-sided hypergeometric test using TBtools software[70]. **c** Upregulation of salicylic acid (SA)-related genes in ds*LsSP1*-treated *L. striatellus* infested plants compared with ds*GFP*-treated *L. striatellus* infested ones. **d** H₂O₂ levels in the untreated rice plants and rice

plants infested by dsRNA-treated *L. striatellus*. Different lowercase letters indicate statistically significant differences at *P* < 0.05 level according to one-way ANOVA test followed by Tukey's multiple comparisons test. **e** Upregulation of defense genes in ds*LsSP1*-treated *L. striatellus* infested plants compared with ds*GFP*-treated *L. striatellus* infested ones. *P*-values were determined by two-tailed unpaired Student's *t* test. **P* < 0.05; ***P* < 0.01; ****P* < 0.001; ns, not significant. PAD4 phytoalexin deficient 4, SAMT SA methyl transferase, SAGT SA glucosyl transferase, WRKY transcription factor WRKY, PR1 pathogenesis-related 1. Data in **d** and **e** are presented as mean values ±SEM (*n* = 3 biological replicates). The rice variety cv. ASD7 was used. Source data are provided as a Source Data file.

genes (Supplementary Table 2), 8 were found to be differentially expressed. These DEGs were all up-regulated (Fig. 4c), indicating the activation of SA pathway in ds*LsSP1*-treated *L. striatellus* infested plants compared with ds*GFP*-treated *L. striatellus* infested ones. $H_2O_2$ accumulation has been applied as a marker for plant basal defenses against sap-sucking herbivores[29,35]. In this study, $H_2O_2$ levels in rice plants were significantly higher at 24 h after ds*LsSP1*-treated *L. striatellus* infestation than those after ds*GFP*-treated *L. striatellus* infestation (one-way ANOVA test followed by Tukey's multiple comparisons test, $p = 0.0298$; Fig. 4d). Quantitative real-time PCR (qRT-PCR) analysis further confirmed the upregulation of defense genes (Fig. 4e), and the obtained results were consistent with transcriptomic data. Collectively, these results demonstrate that a deficiency in LsSP1 secretion activates plant defenses as a response to *L. striatellus* infestation.

### Overexpressing LsSP1 in rice plants benefits ds*LsSP1*-treated *L. striatellus* feeding

Transgenic Nipponbare rice plants with constitutive LsSP1 overexpression were constructed (Supplementary Fig. 17). The wild-type (WT) Nipponbare plant was used as a control. Two independent homozygous lines were used, and similar results were obtained. The results of comparison group 1 (WT and *oeSP1#1*) and comparison group 2 (WT and *oeSP1#2*) are presented in Fig. 5 and Supplementary Fig. 18, respectively. The resistance of transgenic plants to *L. striatellus* (4th instar; wild-type) infestation was firstly investigated. No significant resistance changes in *oeSP1* plants were found when compared with WT plants (two-tailed unpaired Student's *t* test, $p = 0.4880$ in comparison group 1, $p = 0.6704$ in comparison group 2; Supplementary Fig. 19). Compared with ds*GFP*-treated controls, the treatment of *L. striatellus* with ds*LsSP1* did not affect insect survivorship after feeding on *oeSP1* plants (log-rank test, $p = 0.9913$ on *oeSP1#1*, $p = 0.5715$ on *oeSP1#2*; Supplementary Fig. 20). For fecundity analysis, the ds*LsSP1*-treated *L. striatellus* produced less offspring than the ds*GFP*-treated control when feeding on WT plants (two-tailed unpaired Student's *t* test, $p = 0.0299$; Fig. 5a). However, this detrimental effect was not observed when ds*LsSP1*-treated *L. striatellus* fed on *oeSP1* plants (two-tailed unpaired Student's *t* test, $p = 0.8771$ in comparison group 1, $p = 0.7670$ in comparison group 2; Fig. 5a and Supplementary Fig. 18a). For honeydew excretion, the ds*LsSP1*-treated *L. striatellus* excreted less honeydew than the ds*GFP*-treated control when feeding on WT plants, although with no statistical significance (two-tailed unpaired Student's *t* test, $p = 0.1047$; Fig. 5b). There was also no significant difference in honeydew excretion between ds*GFP*- and ds*LsSP1*-treated *L. striatellus* when feeding on *oeSP1* plants (two-tailed unpaired Student's *t* test, $p = 0.3751$ in comparison group 1, $p = 0.9523$ in comparison group 2; Fig. 5b and Supplementary Fig. 18b). EPG was subsequently used to monitor the insect feeding behavior on transgenic plants. Compared with ds*GFP*-treated controls, the ds*LsSP1*-treated *L. striatellus* exhibited a significant decrease in phloem sap ingestion when feeding on WT plants (two-tailed unpaired Student's *t* test, $p = 0.0426$ in comparison group 1, $p = 0.0037$ in comparison group 2; Fig. 5c and Supplementary Fig. 18c). Nevertheless, no significant difference in phloem sap ingestion was observed between ds*GFP*- and ds*LsSP1*-treated *L. striatellus* feeding on *oeSP1* plants (two-tailed unpaired Student's *t* test, $p = 0.8913$ in comparison group 1, $p = 0.9390$ in comparison group 2; Fig. 5c and Supplementary Fig. 18c), indicating that overexpression of LsSP1 in rice plants rescued the feeding defects caused by a deficiency in LsSP1 secretion.

To comprehensively illustrate the effects of LsSP1 on rice plants, transcriptomic analyses were performed on WT and *oeSP1#1* plants that were untreated or infested by ds*LsSP1*-treated *L. striatellus*. DEGs between untreated and ds*LsSP1*-treated *L. striatellus* infested plants were compared, and totally 3396 and 1998 genes were identified in WT and *oeSP1#1* plants, respectively (Supplementary Data 3–4). There were 2335 DEGs specifically identified in WT plants, but not in *oeSP1#1*

plants, and they were potentially correlated with LsSP1-associated responses. Enrichment analysis revealed that the majority of these genes were involved in plant hormone signal transduction, plant-pathogen interaction, MAPK signal transduction, and amino acid metabolism (Fig. 5d). Among the 28 SA-related genes, 18 were differentially expressed in at least one comparison group, and 16 were significantly up-regulated after infestation (Fig. 5e). Interestingly, these up-regulated genes were induced to a lower extent in *oeSP1#1* plants compared with those in WT plants (Fig. 5e), indicating that LsSP1 overexpression attenuated the *L. striatellus*-induced SA biosynthesis and SA response.

## Discussion

Herbivorous insects have developed dynamic and complex interactions with host plants. Advanced understanding towards their underlying mechanisms will provide the fundamental knowledge for developing efficient pest management strategies. In this study, the role of salivary LsSP1 in its interaction with rice hosts was investigated. Using Y2H, BiFC and LUC assays, we showed that LsSP1 was secreted into plant tissues during feeding and directly interacted with the salivary sheath protein LsMLP. In yeast and *N. benthamiana*, LsSP1 interacted with multiple PLCPs in various subfamilies. LsSP1 knockdown led to a decrease in insect feeding and reduced insect reproduction on WT plants, but not on *oeSP1* plants. Our results indicate that the salivary sheath protein LsSP1, although not essential for salivary sheath formation, is beneficial for insect performance.

During the feeding process, herbivorous insects can secrete hundreds of proteins into plant tissues. Previously, most salivary proteins are investigated individually, and it is found that different salivary proteins from one species exerted diversified roles in insect-plant interactions[36,37]. For example, in *M. persicae*, the overexpression of salivary protein Mp10 activates multiple defense pathways in *N. benthamiana* plants and reduces aphid performance[36,38]. However, the overexpression of another salivary protein Mp55 increases the attraction of *N. benthamiana* plants to aphids, and promotes aphid performance[37]. *L. striatellus* can successfully ingest rice phloem saps with limited plant defenses. However, when LsMLP was overexpressed, elevated accumulation of $H_2O_2$ was detected (Supplementary Fig. 15), which was in contradiction with the actual feeding situation. Therefore, there must exist other salivary components responsible for attenuating the LsMLP-induced plant defenses or masking the LsMLP signal. To the best of our knowledge, no such case has been reported in this insect species, although several proteins in aphids and mirid bug are found to be capable of inhibiting the plant defenses triggered by bacterial flag22 or oomycete INF1[38–40]. Our study demonstrated that LsSP1 bound to LsMLP directly, providing clues that LsSP1 may prevent the activation of plant defenses by masking the LsMLP, which deserves further investigation.

Apoplastic PLCPs act in the front line of plant immunity against a wide range of pathogens, including fungi, bacteria, and oomycetes[41]. Depletion or knockdown of proteases such as Rcr3, RD19, and Pip1 significantly decreases the plant susceptibility to the invading pathogens[42–44]. In maize, PLCPs are required to release the bioactive Zip1, a small peptide that activates SA signaling[45]. In turn, Zip1 release will enhance PLCP activity, thereby establishing a positive feedback loop and promoting the SA-mediated defenses[45]. Our study demonstrated that rice genes related to SA signaling were differentially expressed when infested by *L. striatellus*, and OsOryzain was significantly induced upon SA treatment and *L. striatellus* infestation (Supplementary Figs. 11 and 12). This result might be an indicator that OsOryzain is regulated through SA pathway. SA signaling plays an important role in the rice defense against planthoppers[33]. The transcript level of *OsOryzain* reached a peak at 6 h-post *L. striatellus* infestation, while a peak was reached at 12 h-post SA treatment (Supplementary Fig. 11a). The different induction patterns indicated that

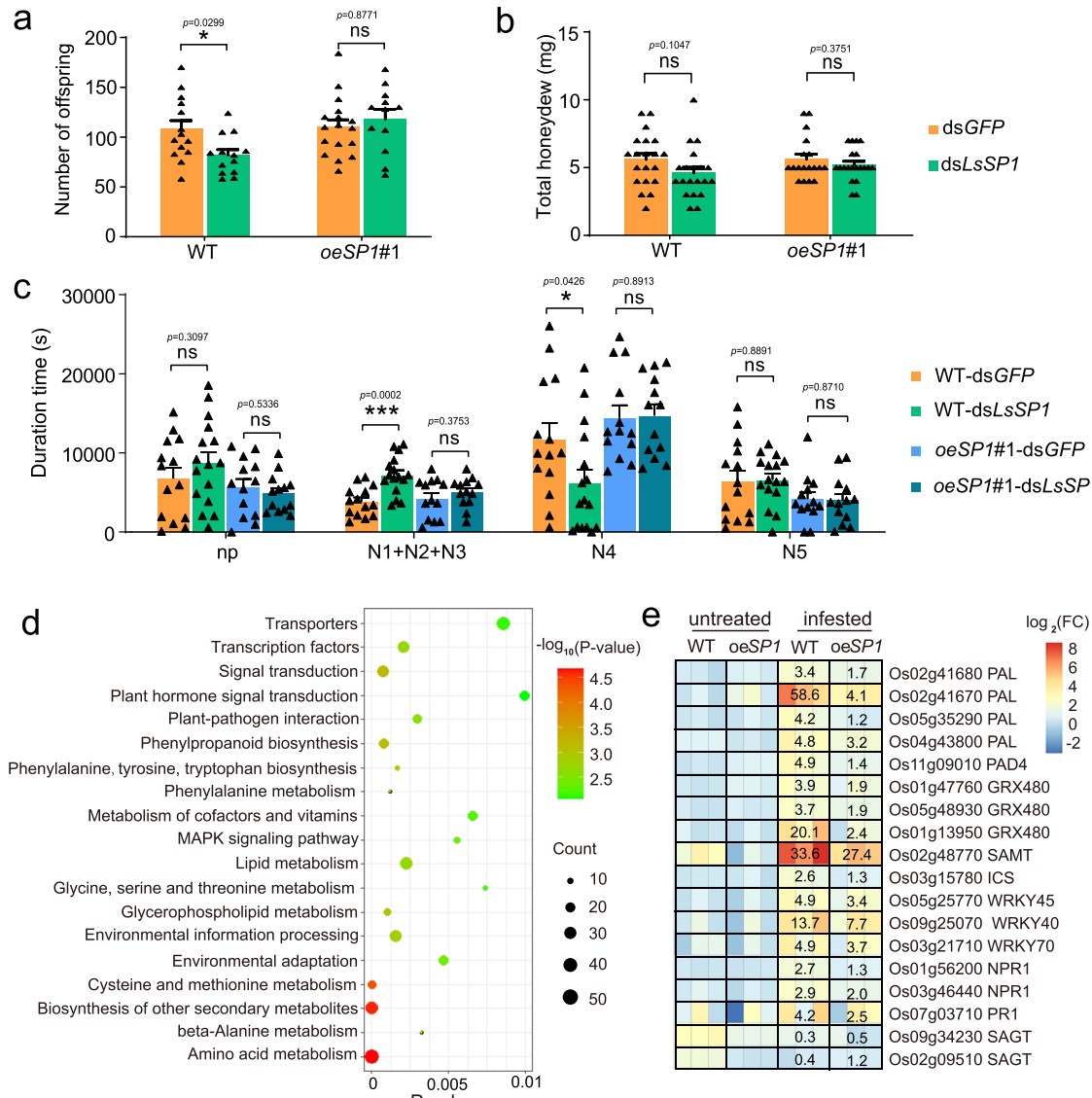

**Fig. 5 | Influences of *Laodelphax striatellus* infestation on wild type (WT) and *oeSP1*#1 plants. a**–**c** Comparison of fecundity (**a**), honeydew excretion (**b**), and electrical penetration graph (EPG) parameters (**c**) between ds*GFP*-treated and ds*LsSP1*-treated *L. striatellus* on WT and *oeSP1*#1 plants. *P*-values were determined by two-tailed unpaired Student's *t* test. **P* < 0.05; ***P* < 0.01; ****P* < 0.001; ns, not significant. Data are presented as mean values ±SEM. For fecundity (**a**) and honeydew (**b**) analysis, *n* = 20 independent biological replicates in each treatment; for EPG analysis (**c**), *n* = 14 (WT-ds*GFP*), *n* = 16 (WT-ds*LsSP1*), *n* = 13 (*oeSP1*#1-ds*GFP*), and *n* = 13 (*oeSP1*#1-ds*LsSP1*) independent biological replicates. All EPG recordings were performed for 8 h. N1 + N2 + N3 pathway duration, N4 phloem sap ingestion, N5 xylem sap ingestion, np nonpenetration. **d** Kyoto Encyclopedia of Genes and

Genomes (KEGG) pathway enrichment analysis of DEGs specifically identified in WT plants, but not in *oeSP1*#1 plants. Enriched *P*-values were calculated according to one-sided hypergeometric test using TBtools software[70]. **e** The impacts of *L. striatellus* infestation on salicylic acid (SA) marker genes in WT and *oeSP1*#1 plants. PAL phenylalanine ammonia lyase, PAD4 phytoalexin deficient 4, GRX480 glutaredoxin GRX480, SAMT SA methyl transferase, ICS isochorismate synthase, WRKY transcription factor WRKY, NPR1 non-repressor of pathogenesis-related protein 1, PR1 pathogenesis-related 1, SAGT SA glucosyl transferase. Nipponbare rice plants and transgenic plants of Nipponbare background were used. Source data are provided as a Source Data file.

other factors, in addition to SA pathway, might also be responsible for *OsOryzain* expression, which deserves further investigation.

Although our study showed interaction between LsSP1 and OsOryzain in Y2H assays (Supplementary Note 1 and Supplementary Fig. 7), rice plants knockout of OsOryzain cannot well rescue the feeding defects caused by a deficiency in LsSP1 secretion as that LsSP1 overexpressing plants did (Supplementary Note 5, Fig. 5, and Supplementary Fig. 21). This might be explained by complex interactions between effectors and different plant defense actors. As a case in *Phytophthora*, the multifunctional effector Avrblb2 can neutralize host defense proteases via targeting PLCPs[28], and suppresses defense associated Ca$^{2+}$ signaling pathway by interacting with host calmodulin[46]. For salivary LsSP1, it targets multiple PLCPs belonging to

different subfamilies (Supplementary Fig. 8). The knockout of OsOryzain alone cannot inhibit plant defenses initiated by other PLCPs. In addition, LsSP1 is capable of interacting with other plant and insect proteins (Fig. 3 and Supplementary Table 1). The salivary LsSP1 potentially exerts multiple roles during insect feeding, and affects plant defense in other ways independent of PLCPs, which deserves further investigation.

## Methods

### Insects and plants

The *L. striatellus* strain was originally collected from a rice field in Ningbo China. The insects and rice plants were maintained in a climate chamber at 25 ± 1 °C, with 70–80% relative humidity, and a light/dark

photoperiod of 16/8 h. Two rice varieties (cv. ASD7 and Nipponbare) were used in this study. The resistant variety ASD7, which contained the brown planthopper resistance gene BPH2, was also reported to confer resistance to small brown planthopper[47,48], and was extensively applied for insect bioassays. As the transgenic rice plants generated in this study were of Nipponbare background, wild-type Nipponbare plants were used as a control. Therefore, the rice variety used in transgenic rice plant analysis was Nipponbare. For the rest of rice-associated experiments, ASD7 plants were used. In addition, *N. benthamiana* plants were kept in a growth chamber at $23 \pm 1\,°C$ under a light/dark photoperiod of 16 h/8 h.

### Analysis of genes abundantly expressed in salivary glands

The top 100 genes that were abundantly expressed in *L. striatellus* salivary glands were reported in our previous study[30]. To identify the potential planthopper-specific genes, these 100 genes were first subject to BLAST search against the predicted proteins in *Acyrthosiphon pisum*;[49] *Bemisia tabaci*;[50] *Riptortus pedestris*;[51] *Homalodisca vitripennis*[52], and *Drosophila melanogaster*[53], with a cutoff E-value of $10^{-5}$, respectively. Genes with no homology in the above species were subsequently searched against the NCBI nr database. Only genes with distributions restricted to three planthoppers (*L. striatellus*, *S. furcifera*, and *N. lugens*) were defined as the planthopper-specific genes. Thereafter, the expression patterns of top 100 genes in different tissues were investigated based on the transcripts per million (TPM) expression values. The TPM expression values of *L. striatellus* genes were generated by analyzing the transcriptomic data of salivary gland, gut, fat body, carcass, testis, and ovary, and used in our laboratory to preliminarily investigate the gene expression patterns. The TPM expression values of the top 100 genes were displayed in Supplementary Data 1. For the identification of salivary gland-specific genes, TPM of each gene in salivary gland was compared with that in the other five tissues, respectively. Afterwards, the gene-relative abundance (ratio) in each comparison group was calculated. Genes with fold changes>10 in all comparison groups were considered to be salivary gland-specific genes.

### Sequence analysis

The SignalP 5.0 Server (https://services.healthtech.dtu.dk) was adopted for predicting the presence of signal peptides and cleavage sites. Transmembrane domains were predicted by the TMHMM Server v2.0 (http://www.cbs.dtu.dk/services/TMHMM/). Protein conserved domains were predicted by InterPro (http://www.ebi.ac.uk/interpro/). Best-matched homologs of planthopper species were aligned with the ClustalX program v1.81[54].

### *L. striatellus* infestation and SA treatment

To investigate the effect of *L. striatellus* and SA on rice defense, the 4-5-leaf stage rice seedlings were sprayed with 0.5 µM SA (#84210, Sigma-Aldrich, St. Louis, MO, USA) or infested by 5th instar *L. striatellus* nymph (5 nymphs per plant, confined in a 5-cm plant stems with a plastic cup). The treated plants were maintained in a climate chamber at 25 °C, and samples were collected at indicated time points.

### Quantitative real-time PCR analysis

Different tissue samples from carcasses (20), fat bodies (50), guts (50), and salivary glands (80) were dissected from the 5th instar nymphs in a phosphate-buffered saline (PBS) solution (137 mM NaCl, 2.68 mM KCl, 8.1 mM $Na_2HPO_4$ and 1.47 mM $KH_2PO_4$ at pH 7.4) using a pair of forceps (Ideal-Tek, Switzerland). Similarly, testes (50) and ovaries (20) were collected from adult male and female *L. striatellus*, respectively. The number of insects in each sample was given in the parentheses above. To extract RNA from *N. benthamiana* and rice, plants were firstly grinded with liquid nitrogen. Then, samples were homogenized in the

TRIzol Total RNA Isolation Kit (#9109, Takara, Dalian, China), and total RNA was extracted following the manufacturer's protocols. Afterwards, the first strand cDNA was reverse-transcribed from RNA using HiScript II Q RT SuperMix (#R212-01, Vazyme, Nanjing, China). qRT-PCR was subsequently run on a Roche Light Cycler® 480 Real-Time PCR System (Roche Diagnostics, Mannheim, Germany) using the SYBR Green Supermix Kit (#11202ES08, Yeasen, Shanghai, China). The PCR procedure was as follows, denaturation for 5 min at 95 °C, followed by 40 cycles at 95 °C for 10 s and 60 °C for 30 s. The primers used in qRT-PCR were designed using Primer Premier v6.0 (Supplementary Table 3). *L. striatellus* actin, *O. sativa* actin, and *N. benthamiana* actin were used as internal controls, respectively. The relative quantitative method ($2^{-\Delta\Delta Ct}$) was employed to evaluate the quantitative variation. qRT-PCR results with a Ct value ≥35 were regarded that the gene was not expressed in the sample. Three independent biological replicates with each repeated twice were performed.

### RNA interference

The DNA sequences of target genes were amplified using the primers listed in Supplementary Table 3, and cloned into pClone007 Vector (#TSV-007, Tsingke, Beijing, China). The PCR-generated DNA templates containing T7 sequence was used to synthesize the double-stranded RNAs with a T7 High Yield RNA Transcription Kit (#TR101-01, Vazyme). RNA interference experiment was conducted as previously described[55]. Briefly, insects were anaesthetized with carbon dioxide for 5–10 s. Then, dsRNA was injected into the insect mesothorax using FemtoJet (Eppendorf-Netheler-Hinz, Hamburg, Germany). Afterwards, insects were kept on the 4–5-leaf stage rice seedlings for 24 h and the living insects were selected for further investigation. Silencing efficiency was determined at 4th day post-injection using qRT-PCR method as described above.

### Insect bioassays

To perform survivorship analysis, a group of 30–40 treated insects (3rd instar nymph) were treated with dsRNA and kept on 4–5 leaf stage rice seedlings in a climate chamber. The mortality rates for each treatment were recorded for ten consecutive days. Three independent replications were performed. For honeydew analysis, a parafilm (Bemis NA, Neenah, WI, USA) sachet was attached to the host plant stems, and the insects (5th instar nymph) were confined in a sachet. At 24 h after feeding, the accumulation of honeydew was measured by weighing the parafilm sachet before and after feeding with an electronic balance (accuracy, 0.001 g; Sartorius, Beijing, China). At least 10 replicates were performed for each treatment. For fecundity analysis, the newly emerged adults were treated with dsRNA. One day later, the insects were paired and allowed for oviposition for 10 days. Afterwards, the number of hatched offspring was counted. At least 10 replicates were conducted for each treatment.

### Host choice test

The 4–5-leaf stage rice seedling was first placed in a glass tube, and 5 dsRNA-treated *L. striatellus* (5th instar) were allowed to feed on one rice plant for 24 h. Thereafter, the insects were removed, and rice plants pre-infested by different dsRNA-treated *L. striatellus* were confined in a plastic cup (diameter, 6 cm; height, 10 cm), where a release chamber was contained. Later, a group of 17 *L. striatellus* (4th instar; wild-type, WT) were placed in the release chamber. The numbers of insects settling on each plant were counted at 1, 3, 6, 12, 24, 36, and 48 h, respectively. At least ten replicates were performed.

### EPG recording analysis

The GiGA-8d EPG amplifier (Wageningen Agricultural University, Wageningen, The Netherlands) with a 10 TΩ input resistance and an input bias current less than 1 pA was used for EPG recording. Briefly, the dsRNA-treated *L. striatellus* (5th instar) were reared on filter paper

with only water provided for 12 h. After anesthetizing by $CO_2$ for 10 s, a gold wire (Wageningen Agricultural University, diameter, 20 mm; length, 5 cm) was applied in connecting insect abdomen and the EPG amplifier with a water-soluble silver conductive glue (Wageningen Agricultural University). The plant electrode was designed by inserting a copper wire (diameter, 2 mm; length, 10 cm) into soils that were planted with one rice plant. Later, EPG recording was conducted for 8 h in a Faraday cage (120 cm × 75 cm × 67 cm, Dianjiang, Shanghai, China), with the gain of the amplifier being set at 50× and the output voltage being adjusted between −5V and +5 V.

The output data were analyzed by PROBE 3.4 (Wageningen Agricultural University), and the insect feeding behaviors were classified into nonpenetration (np), pathway duration (N1 + N2 + N3), phloem sap ingestion (N4), and xylem sap ingestion (N5)[48]. At least 10 replicates were performed for each treatment.

## Immunohistochemistry staining

To prepare insect tissues, salivary glands were dissected from *L. striatellus* and fixed in 4% paraformaldehyde (#E672002, Sango Biotechnology, Shanghai, China) for 30 min. To prepare salivary sheath sample, the parafilm attached with salivary sheath was washed in PBS and fixed in 4% paraformaldehyde for 30 min. To prepare plant tissues, the rice plants infested by *L. striatellus* were collected and cut into segments ~3 cm in length using a scalpel. Then, the short rice sheaths were fixed in 4% paraformaldehyde and vacuumized at 4 °C. Afterwards, the sheaths were blocked with Jung Tissue Freezing Medium (#020108926, Leica Microsystems, Wetzlar, Germany) at −40 °C. Later, the blocks were cut into 20 μm cross-sections using Cryostar NX50 (Thermo Scientific, Waltham, MA), and fixed in 4% paraformaldehyde for the additional 30 min. The anti-LsSP1 serum, prepared by immunizing rabbits with purified GST-LsSP1 proteins, was produced via the custom service of Huaan Biotechnology Company (Hangzhou, China). The anti-OsOryzain serum, prepared by immunizing rabbits with peptides VRMERNIKASSGKC and DVNRKNAKVVTIDSY, was produced via the custom service of Genscript Biotechnology Company (Nanjing, China). The anti-LsSP1 serum was conjugated with Alexa Fluor™ 488 NHS Ester (#A20000, ThermoFisher Scientific), while the anti-OsOryzain serum was conjugated with Alexa Fluor™ 555 NHS Ester (#A37571, ThermoFisher Scientific) following the manufacturer's protocols. Thereafter, the insect/plant/parafilm samples were incubated with the above fluorophore-conjugated serums overnight at 4 °C with a dilution of 1:200, the actin dye phalloidinrhodamine (#A22287, ThermoFisher Scientific) at room temperature with a dilution of 1:500 for 30 min, and 4′,6-diamidino-2-phenylindole (DAPI) solution (#ab104139, Abcam, Cambridge, USA). Finally, fluorescence images were obtained using a Leica confocal laser-scanning microscope SP8 (Leica Microsystems).

## Preparation of protein samples

Salivary sheath samples and watery saliva samples were collected from 900 to 1000 nymphs as previously described[14,56]. Briefly, the 5th instar *L. striatellus* nymphs were transferred from the rice seedlings into a plastic Petri plate. Approximately 300 μl diets with 2.5% sucrose were added between two layers of stretched Parafilm, and the insects were allowed to feed for 24 h. Ten devices were used for saliva collection, with each device containing 90–100 *L. striatellus*. For the preparation of watery saliva samples, the liquid was collected from the space between two layers of Parafilm. To prepare salivary sheath samples, the upper surface of Parafilm with salivary sheath firmly attached was carefully detached, and washed in PBS thrice. As salivary sheath was difficult to dissolve, a lysis buffer of 4% 3-[(3-cholamidopropyl)-dimethylammonio]-1-propanesulfonate (#20102ES03, Yeasen), 2% SDS (#A600485, Sango Biotechnology) and 2% DTT (#A100281, Sango Biotechnology) was adopted for obtaining the solubilized salivary sheath proteins under gentle shaking on an orbital shaker at room

temperature for 1 h according to previous description[14,56]. With this method, the majority of salivary sheath, although not all, can be dissolved[56]. Since it was difficult to quantify the protein concentration in saliva solution, the salivary sheath samples and watery saliva samples was pooled to 50 μl using 3-kDa molecular-weight cutoff Amicon Ultra-4 Centrifugal Filter Device (Millipore, MA, USA), respectively.

The rice apoplast was collected with Buffer A (consisting of 0.1 mol/L Tris-HCl, 0.2 mol/L KCl, 1 mmol/L PMSF, pH 7.6) as previously described[57]. Briefly, 5.0 g rice plants were vacuum infiltrated with Buffer A for 15 min. Then, the remaining liquid on the surface was dried with the absorbent paper, placed inside the 1-ml tips and centrifuged in the 50-ml conical tubes at $1000 \times g$ for 20 min. The apoplastic solution was concentrated using 3-kDa molecular-weight cutoff Amicon Ultra-4 Centrifugal Filter Device.

For the preparation of insect and plant samples, the insects/plants were collected at indicated time points and homogenized in the RIPA Lysis Buffer (#89900, ThermoFisher Scientific). To detect the secretion of LsSP1 into rice plants, approximately one hundred 5th instar nymphs were confined in the 2-cm stem and allowed to feed for 24 h. The outer rice sheath was collected for western-blotting assay.

## Western-blotting assay

The protein concentrations were quantified using a BCA Protein Assay Kit (#CW0014S, CwBiotech, Taizhou, China) in line with the manufacturer's instructions. After the addition of 6× SDS loading buffer, the protein samples were boiled for 10 min. Proteins were separated by 12.5% SDS-PAGE gels and transferred to PVDF membranes. Then, the blots were probed with anti-LsSP1 serum or anti-OsOryzain serum diluted at 1:5000, followed by additional incubation with horseradish peroxidase (HRP)-conjugated goat anti-rabbit IgG antibody (1:10,000, #31460, ThermoFisher Scientific). Images were acquired by an AI 680 image analyzer (Amersham Pharmacia Biotech, Buckinghamshire, UK). The band intensities in immunoblot analyses were quantified using ImageJ software v1.53e (https://imagej.nih.gov/). To monitor the equal protein loading, samples were further stained with Coomassie brilliant blue (CBB). The full scan results of blots and gels were provided in Supplementary Fig. 23 and Source Data file.

## Identification and phylogenetic analysis of PLCP

The *O. sativa* PLCPs were investigated based on the procedure described previously[25]. Briefly, amino acid sequences of 31 *Arabidopsis thaliana* PLCPs[58] were retrieved and used as queries to search for PLCP homologs in the Rice Genome Annotation Project Database (http://rice.plantbiology.msu.edu), with a cutoff e-value of $10^{-5}$. The putative PLCPs were further validated by aligning to the NCBI nr database. Thereafter, the structure and conserved domains of PLCPs were analyzed by InterPro. Seven proteins predicted in Rice Genome Annotation Project Database were incomplete, including Os04g55650 (NP_001389372), Os09g39160 (BAD46641), Os09g39090 (XP_015611357), Os09g39170 (BAD46642), Os09g39120 (XP_015611254), Os01g24570 (BAD53944), and Os07g01800 (BAC06931). The complete sequences were retrieved from the NCBI database by BLAST search, and the corresponding GenBank accessions were provided in the brackets. For phylogenetic analysis, all PLCPs were aligned with MAFFT v7.450, and the gaps were further trimmed using Gblock v0.91b[59]. The substitution model was evaluated using ModelTest-NG based on the default parameters[60]. Afterwards, maximum likelihood (ML) trees were constructed using RAxMLNG v0.9.0 with 1000 bootstrap replications[61].

## Scanning electron microscopy

Insects were allowed to feed on rice plants or artificial diets for 24 h. The rice plant and parafilm attached with salivary sheath were cut and washed with PBS. Later, SEM samples were attached to a stub and dried in a desiccator under vacuum. After gold-sputtering, the samples were observed by SEM TM4000 II plus (Hitachi, Tokyo, Japan). The length of

salivary sheath on the Parafilm membrane was measured from the top to base of salivary sheath (Supplementary Fig. 4a), while the number of salivary sheaths left on the rice surface was measured by counting the ring-shaped salivary sheath structure (Supplementary Fig. 4c) on 4-cm rice stem.

### Agrobacterium-mediated plant transformation and diaminobenzidine staining

Details in Agrobacterium-mediated plant transformation in *N. benthamiana* and diaminobenzidine staining of *N. benthamiana* leaves were described in Supplementary Methods.

### Protein−protein interaction assays

Details in the Y2H screening assay, Y2H point-to-point verification assay, GST pull-down assay, BiFC assay, luciferase complementation (LUC) assay, and OsOryzain-salivary sheath binding assay were described in Supplementary Methods.

### Generation of transgenic rice plants

To generate the *oeSP1* plants, the coding sequence (without signal peptide) was amplified and cloned into the binary expression vector driven by a CaMV 35S promoter. The recombinant vector was introduced into *A. tumefaciens* strain EHA105 by the heat transfer method. Transgenic rice plants were generated through *Agrobacterium*-mediated transformation. Briefly, rice seeds (cv. Nipponbare) were sterilized with 75% ethanol for 1 min and 50% sodium hypochlorite for 20 min. After washing in sterile water for three times, the sterilized seeds were transferred onto NBi medium (N6 macro elements, B5 microelements, B5 vitamin, 27.8 mg/L $FeSO_4 \cdot 7H_2O$, 37.3 mg/L $Na_2$-EDTA, 500 mg/L proline and glutamic acid, 300 mg/L casein hydrolyte, 2 mg/L 2,4-dichlorophenoxyacetic acid, 100 mg/L inositol, and 30 g/L sucrose) for 20 days at 26 °C for callus induction. The induced calli were incubated with *Agrobacterium* ($OD_{600} = 0.2$) for 10 min, and then cultured in NBco medium (NBi medium supplied with 100 μmol/L acetosyringone, pH 5.5) for 3 days at 20 °C. After washing with sterile water, the calli were transferred onto NBs medium (NBi medium supplied with 500 mg/L cephamycin and 30 mg/L hygromycin) for 25 days. Subsequently, the resistant calli were transferred onto NBr medium (NBi medium supplied with 0.5 mg/L α-naphthalene acetic acid, 3 mg/L 6-benzylaminopurine, 500 mg/L cephamycin, and 30 mg/L hygromycin) for shoot regeneration. The regenerated shoots were transferred into 1/2× Murashige−Skoog medium for rooting. The transgenic plants were grown in the greenhouse, and was confirmed by RT-PCR with reverse-transcribed cDNA as the template using LsSP1-specific primers (Supplementary Table 3). Two independent T3 homozygous over-expression lines (Supplementary Fig. 17a) were used for subsequent experiments.

### Evaluation of *L. striatellus* resistance in transgenic rice plants

The *L. striatellus* resistance in rice plants was scored as previously described[62,63]. Briefly, five rice seedlings were grown in a 10-cm-diameter plastic cup with a hole at the bottom. At the 4−5 leaf stage, the seedlings were infested with *L. striatellus* nymphs (4th instar; wild-type, WT) at a dose of 10 insects per seedling. After 20 days, the injury level of rice plants was checked, and the identification standard was adopted for calculating the average injury level[62] (Supplementary Table 4). Four replicates were performed for each line.

### Performance of dsRNA-treated *L. striatellus* on transgenic rice plants

To investigate the performance of dsRNA-treated *L. striatellus* on transgenic rice plants, 3rd instar nymph (for survivorship analysis), 4th instar nymph (for honeydew and EPG analyses), and newly emerged adults (for fecundity analysis) were treated with ds*GFP* and ds*LsSP1*,

respectively. Insect bioassays for survivorship, honeydew, fecundity, and EPG analyses were performed as described above. Two independent homozygous overexpression/knockout transgenic lines were used.

### Transcriptomic sequencing

The untreated rice plants or rice plants infested by ds*LsSP1*-treated *L. striatellus* for 24 h were collected and homogenized in the TRIzol Reagent (#10296018, Invitrogen, Carlsbad, CA, USA). Thereafter, total RNA was extracted according to the manufacturer's instructions, and the RNA samples were sent to Novogene Institute (Novogene, Beijing, China) for transcriptomic sequencing as previously described[64]. Briefly, poly(A) + RNA was purified from 20 μg pooled total RNA by using oligo(dT) magnetic beads. Fragmentation was implemented in the presence of divalent cations at 94 °C for 5 min. Then, N6 random primers were used for reverse transcription into the double-stranded complementary DNA (cDNA). After end-repair and adapter ligation, the products were amplified by PCR and purified using a QIAquick PCR purification kit (Qiagen, Hilden, Germany) to create a cDNA library. The library was sequenced on an Illumina NovaSeq 6000 platform. Thereafter, all sequencing data generated were submitted to the NCBI Sequence Read Archive under accession number PRJNA833487 and PRJNA815455.

### Analysis of transcriptomic data

The output raw reads were filtered using the internal software, and the clean reads from each cDNA library were aligned to the reference sequences in Rice Genome Annotation Project Database using HISAT v2.1.0[65]. The low-quality alignments were filtered by SAMtools v1.7[66]. Transcripts per million (TPM) expression values were calculated using Cufflink v2.2.1[67]. The DESeq2 v2.2.1[68] was adopted for analyzing the DEGs, and genes with log2-ratio > 1 and adjusted $p$ value < 0.05 were identified. To reveal overall differences in gene expression patterns among different transcriptomes, R function plotPCA (github.com/franco-ye/TestRepository/blob/main/PCA_by_deseq2.R) and DNAstar v8.0[69] were used to perform PCA analysis and correlation analysis, respectively. KEGG enrichment analyses were performed using TBtools software v1.0697[70]. In this software, enriched *P*-values were calculated according to one-sided hypergeometric test: $P = 1 - \sum_{i=0}^{m-1} \left( \frac{\binom{M}{i}\binom{N-M}{n-i}}{\binom{N}{n}} \right)$, with $N$ represents the number of gene with KEGG annotation, $n$ represents the number of DEGs in N, M represents the number of genes in each KEGG term, $m$ represents the number of DEGs in each KEGG term.

### Statistical analysis

The log-rank test (SPSS Statistics 19, Chicago, IL, USA) was applied to determine the statistical significance of survival distributions. Two-tailed unpaired Student's *t* test (comparisons between two groups) or one-way ANOVA test followed by Tukey's multiple comparisons test (comparisons among three groups) was used to analyze the results of qRT-PCR, EPG, proteolytic activity, honeydew measurement, offspring measurement, and host choice analysis. The exact *p* value of each statistical test was provided in Source data file. Data were graphed in GraphPad Prism 9.

### Reporting summary

Further information on research design is available in the Nature Portfolio Reporting Summary linked to this article.

## Data availability

The sequencing data generated in this study have been deposited in the NCBI Sequence Read Archive under accession number PRJNA833487 and PRJNA815455. The TPM expression values of all

genes generated from sequencing data can be found in Source data file. Sequence data can be found in GenBank under the following accession numbers: LsSP1, ON322955; NlSP1, ASL05017; SfSP1, ON322954; OsOryzain, NP_001389372.1 [https://www.ncbi.nlm.nih.gov/protein/NP_001389372]; LsMLP, ON568348; NlMLP, KY348750; SfMLP, AQP26312; hypothetical protein DAI22 06g016200, KAF2924946.1 [https://www.ncbi.nlm.nih.gov/protein/KAF2924946]; putative receptor-like protein kinase, XM_015785635; SNF1-related protein kinase regulatory, XP_015639150.1; polypyrimidine tract-binding protein, XP_015632933.1 [https://www.ncbi.nlm.nih.gov/protein/XP_015632933]; alpha-galactosidase, NP_001390973.1 [https://www.ncbi.nlm.nih.gov/protein/NP_001390973]; alpha-amylase, NP_001390734.1 [https://www.ncbi.nlm.nih.gov/protein/NP_001390734]; putative cysteine proteinase Os09g39160, BAD46641 [https://www.ncbi.nlm.nih.gov/protein/BAD46641.1]; zingipain-2 Os09g39090, XP_015611357; putative cysteine proteinase Os09g39170, BAD46642; ervatamin-B Os09g39120, XP_015611254; putative cysteine protease Os01g24570, BAD53944; and Os07g01800, BAC06931. The *O. sativa* reference genome was public available in Phytozome (https://data.jgi.doe.gov/refine-download/phytozome?organism = Osativa&expanded=323). PLCP accessions were listed in Supplementary Fig. 8 and the corresponding sequences can be found in Source data file. Sequences of top 100 genes that abundantly expressed in *L. striatellus* salivary glands can be found in Source data file. Source data are provided with this paper.

## Code availability

The code used in PCA analysis has been deposited in Github: https://github.com/franco-ye/TestRepository/blob/main/PCA_by_deseq2.R.

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

## Acknowledgements

This project has received funding from the National Key Research and Development Plan in the 14th five year plan (2021YFD1401100: H.J.H. and C.X.Z.), the "Pioneer" and "Leading Goose" R&D Program of Zhejiang (2022C02047: H.J.H. and C.X.Z.), Young Elite Scientists Sponsorship Program by CAST (2021QNRC001: H.J.H.), and the National Natural Science Foundation of China (31801734: H.J.H). We thank Shao-Fei Rao for providing ER, Golgi, and tonoplast markers; He-Hong Zhang for providing pCV-cYFP, pCV-nYFP, pCAMBIA1300-cLUC, and pCAMBIA1300-nLUC vectors; Yan-Juan Jiang for the critical comments for the manuscript.

## Author contributions

H.-J.H., J.-M.L., C.-X.Z., and J.-P.C. planned and designed the research. H.-J.H., Y.-Z.W., L.-L.L., H.-B.L., J.-B.L., X.W., Z.-X.Y., Y.-J.H., Z.-L.Z., G.L., and J.-C.Z. performed experiments and analyzed data. Q.-Z.M. and Z.-T.S. provided valuable suggestion for the research. H.-J.H. drafted the manuscript, H.-J.H. and C.-X.Z. revised the manuscript.

## Competing interests

The authors declare no competing interests.
