## [Peer Review File · Nature Communications]

Reviewers' Comments:

Reviewer #1:

Remarks to the Author:

During co-evolution with pathogens and insects, plants have evolved multi-layered immune systems from recognizing invaders to activating defense responses. Plant-insect interactions is becoming an exciting frontier of plant science. The manuscript revealed that the main component of salivary sheath, mucin-like protein LsMLP could trigger rice papain-like cysteine proteases (PLCPs, OsOryzain)-mediated plant defenses. While a salivary sheath protein LsSP1 critical for *L. striatellus* feeding on rice plants could act as an effector that targets multiple PLCPs and suppresses plant defenses triggered by OsOryzain and LsMLP. The manuscript is a good piece of work and written well as well.

Major concern:

According to the model proposed by the authors, LsMLP secreted by *Laodelphax striatellus* was perceived by the host plant, triggering the SA-induced defense responses, including the upregulation and secretion of PLCPs into apoplast, which in turn establishing a positive feedback loop to inhibit insect feeding. LsSP1 could potentially prevent the activation of plant defense by masking the LsMLP and also interact with multiple PLCPs, thus restricting these immune proteases to salivary sheath and inhibiting their proteolytic activities.

In Line 124-126, "Noteworthy, LsSP1 deficiency did not affect salivary sheath formation, and there were no significant differences in salivary sheath appearance, salivary sheath length or the number of salivary sheath left on the rice surface between dsLsSP1 treatment and the control (Fig. S3)".

In Line 216-217, "According to our results, the salivary sheath secreted from dsGFP- and dsLsSP1-treated insects shared very similar morphology".

In Line 225-227, "In comparison, the Alexa Fluor™ 555 signal (OsOryzain protein) in dsLsSP1-infested plants did not cluster around the salivary sheath (Fig. 6C), suggesting that OsOryzain failed to bind to salivary sheath in the absence of LsSP1 in rice plants".

It's a bit difficult to understand such results. LsSP1 deficiency could not suppress salivary elicitor-induced plant defenses, which is mediated by OsOryzain (PLCPs). In this situation, OsOryzain (PLCPs) could function very well.

Minor concerns:

1. In Line 112-113 and Line 118-119, LsSP1 band detected in rice plants infested by *L. striatellus* (Fig. 2C) seemed much stronger than that of in salivary sheath sample (Fig. 2D). It looks unreasonable as saliva secreted into rice plants should be much less than that of collected from artificial diet. Please clarify.

2. Line 153-154. Can the authors provide other LsSP1 candidate interactors revealed by Y2H screening?

3. The interaction between LsSP1 and LsMLP is critical to the authors' finding. Please validate their interaction with more experimental methods such as Co-IP and luciferase complementation assay.

4. Can the authors provide Oryzain gene accession number in GenBank or locus identifier in Rice Genome Annotation Project Database? In Line 849-850, the authors indicated the OsOryzain identified by Y2H screening with arrow, but which can't be found in Figure S5. This could also be applied to LsSP1, although the authors provided nucleic acid and amino acid sequence of LsSP1 in Figure S1.

5. As indicated by Fig 5 A and B, it looked like that *L. striatellus* infestation was more effective to induce the expression of OsOryzain and OsaALP-1 than that of SA treatment. Can the authors explain it in a reasonable way?

6. In Line 745-747, the protein level of OsOryzain in rice sheath (C) in response to SA and *L. striatellus* (L.s.) were determined at 12 and 24 h post treatments (Fig 5C). Did the authors perform the WB assay of these protein samples on the same gel? If not, I am suggesting the authors compare the protein level of OsOryzain in the same gel (or blot). This could also be applied to Fig 5D.

7. Line 273-275, the expression of NbPR1 was not significantly suppressed in the presence of LsSP1-mCherry in Fig 7E, please revise it.

8. Please indicate statistically significant differences in Fig 8A.

Reviewer #2:

Remarks to the Author:

The authors present a very thorough study into how two planthopper proteins interact with the rice immune system, leading to lowered cysteine protease-mediated plant defenses. The design of the study is elegant and the results are exciting. In my opinion, this study constitutes an important advance in the field. However, there are a few aspects of the manuscript that I think could be improved:

Title and throughout the manuscript

Evolution has no purpose, traits do not evolve with a preconceived aim. The title and manuscript text need to undergo rigorous editing to reflect this better.

Abstract

Lines 23-25: The authors use the phrase "fierce arms races" here and elsewhere in the manuscript, but they do not present data on the evolutionary dynamics in this study system. For example, the authors only study a single genotype of the insect and a single genotype of the plant. The authors do not present evidence of arms race dynamics being at play in this system? Arms race dynamics can occur when co-evolution takes place in species interactions, but a lot more work would be required to check if co-evolution is taking place here. See for example the paragraph "Conditions for co-evolution" in the following paper: <https://www.nature.com/articles/ng1202-569>

Introduction

Line 33: The nematodes that the authors refer to are herbivores as well? And which other herbivores do the authors refer to here?

Line 36: There is more than one non-self molecule.

Line 48: These effector proteins were all characterized in the aphid *Myzus persicae*, so the authors might as well mention the species. The effectors do not negatively affect insect reproduction necessarily. This depends on the host plant-insect combination. It would be helpful if the authors could mention the host plant species and genotypes as well. The authors do not introduce what the Mp molecules are either, a reader needs more information for this paragraph to make sense.

Line 49: Which plant species and genotypes can recognize these proteins? And again, the authors have not introduced that these molecules are proteins.

Line 62: Does the salivary sheath contain a single elicitor of plant defense or multiple elicitors?

Lines 63-64: The authors need to be more precise. It has not been shown in the paper that the authors refer to that the salivary sheath evolves faster. That paper only suggests that a few proteins in the salivary sheath may evolve faster.

Results

Line 91-94: Can the authors provide more detail on how they identified LsSP1 as a candidate gene? Has the data that the authors refer to been published previously? Are the proteins specific

to *L. striatellus* or do they also include other planthopper species such as the brown planthopper *Nilaparvata lugens* and the white-backed planthopper *Sogatella furcifera*?

Lines 95-97: It would be helpful for the reader if the authors could provide information on the plants that the insects were reared on.

Line 99: Perhaps use the word "significant" rather than the word "remarkable"?

Lines 155-156: Could the authors provide more detail here? How was the screen performed? Were there proteins other than OsOryzain that were identified?

Line 174: Should it be OsSAG12-1 instead of OsSAG-1? And should the start of the new sentence say "In addition to OsOryzain..." instead of "Besides..."? Is there an explanation why LsSP1 binds to some PLCPs but not others? It looks like the PLCPs it does not bind to do not form a monophyletic clade?

Lines 180-190: Why did the authors decide to focus on OsAALP-1 and not the other PLCPs? Was expression of the other PLCPs tested as well and was expression of these also induced after planthopper feeding and/or SA treatment? Without this information it is impossible for a reader to judge if the expression pattern of OsOryzain is "typical" for a PLCP as mentioned in line 190.

Line 251 and others: It is "mCherry" and not "mCheery"?

Line 275: It appears that expression of NbPR1 is significantly suppressed in the presence of LsSP1-mCherry?

Lines 313-316: This description of the PCA appears too simplistic? oeSP1, and koOry are very similar to one another indeed, but they both are a bit different from WT in both treatment groups?

Discussion

Line 342: The fact that an effector protein suppresses defense in one host, but triggers it in another, does not necessarily indicate that the effector has opposite roles in both hosts, it could just have an opposite effect.

Methods

Line 395: The authors use multiple rice genotypes for their experiments. Which genotypes were used in each experiment?

Line 538: More detail on transcriptome sequencing and analysis if the transcriptome data needs to be provided.

I encourage the authors to proofread the manuscript again for spelling and grammar.

Reviewer #1 (Remarks to the Author):

During co-evolution with pathogens and insects, plants have evolved multi-layered immune systems from recognizing invaders to activating defense responses. Plant-insect interactions is becoming an exciting frontier of plant science. The manuscript revealed that the main component of salivary sheath, mucin-like protein LsMLP could trigger rice papain-like cysteine proteases (PLCPs, OsOryzain)-mediated plant defenses. While a salivary sheath protein LsSP1 critical for *L. striatellus* feeding on rice plants could act as an effector that targets multiple PLCPs and suppresses plant defenses triggered by OsOryzain and LsMLP. The manuscript is a good piece of work and written well as well.

Response: Thank you very much for your comments on our manuscript. We have revised manuscript according to your recommendation.

Major concern:

According to the model proposed by the authors, LsMLP secreted by *Laodelphax striatellus* was perceived by the host plant, triggering the SA-induced defense responses, including the upregulation and secretion of PLCPs into apoplast, which in turn establishing a positive feedback loop to inhibit insect feeding. LsSP1 could potentially prevent the activation of plant defense by masking the LsMLP and also interact with multiple PLCPs, thus restricting these immune proteases to salivary sheath and inhibiting their proteolytic activities.

In Line 124-126, “Noteworthy, LsSP1 deficiency did not affect salivary sheath formation, and there were no significant differences in salivary sheath appearance, salivary sheath length or the number of salivary sheath left on the rice surface between dsLsSP1 treatment and the control (Fig. S3)”.

In Line 216-217, “According to our results, the salivary sheath secreted from dsGFP- and dsLsSP1-treated insects shared very similar morphology”.

In Line 225-227, “In comparison, the Alexa Fluor™ 555 signal (OsOryzain protein) in dsLsSP1-infested plants did not cluster around the salivary sheath (Fig. 6C), suggesting that OsOryzain failed to bind to salivary sheath in the absence of LsSP1 in rice plants”.

It’s a bit difficult to understand such results. LsSP1 deficiency could not suppress salivary elicitor-induced plant defenses, which is mediated by OsOryzain (PLCPs). In this situation, OsOryzain (PLCPs) could function very well.

Response: Thank you for pointing out this and sorry for inadequate description. In previous version, Line 124-126 and Line 216-217 described the same result (from different experiments) that dsLsSP1 treatment did not affect the salivary sheath morphology, including salivary sheath appearance, salivary sheath length or the number of salivary sheath left on the rice surface. **From this result, we concluded that LsSP1 did not influence the formation of salivary sheath**, which is different from two main salivary sheath proteins (NISHP and NIMLP) previously reported. **However, we cannot judge whether LsSP1 deficiency affect plant defenses from this result.** In the revised version, we delete the duplicate description of Line 216-217, and addressed

that “LsSP1 was a salivary sheath protein, but was not indispensable for salivary sheath formation, which was significantly different from two previous reported salivary sheath proteins ^{14, 32}” in Line 134-136.

In Line 225-227, we described that salivary sheath in deficiency of LsSP1 is normal in morphology, but failed to bind OsOryzain in rice plants. From this result and combined with the positive role of OsOryzain in plant immunity, it is high possibility that LsSP1 deficiency could not suppress salivary elicitor-induced, OsOryzain-mediated plant defenses, and **more fierce plant defense might occur in rice infested by dsLsSP1-treated *L. striatellus***, as OsOryzain could function very well in absence of LsSP1. This hypothesis **was validated in comparing rice plants infested by dsGFP- and dsLsSP1-treated *L. striatellus*** from 4 aspects. ¹ Rice plants pre-infested with dsLsSP1-treated *L. striatellus* were less attractive to *L. striatellus* nymphs than those pre-infested with dsGFP-treated *L. striatellus* (Fig. 8A). ² dsLsSP1-treated *L. striatellus* elicited significantly upregulation of rice defense genes involved in plant-pathogen interaction, environmental adaptation, and terpenoids metabolism when compared with dsGFP- treated *L. striatellus* (Fig. 8B). ³ SA pathway, which played critical roles in rice defense against planthoppers, was more fiercely activated in rice plants infested by dsLsSP1-treated *L. striatellus* than that of dsGFP- treated *L. striatellus* (Fig. 8C). ⁴ H₂O₂ accumulation, which was utilized as a marker for plant basal defenses against insects, was significantly higher in rice plants infested by dsLsSP1-treated *L. striatellus* than that of dsGFP- treated *L. striatellus* (Fig. 8E). In the revised version, we addressed that a deficiency in LsSP1 secretion made *L. striatellus* failed to suppress plant defense mediated by PLCPs in Discussion section in Line 397-400: “The deficiency in LsSP1 secretion made *L. striatellus* fail to suppress plant defense mediated by PLCPs (OsOryzain), which led to significant upregulation of defense genes, activation of SA pathway, and accumulation of H₂O₂ in dsLsSP1-infested plants (Fig. 8).”

Minor concerns:

1. In Line 112-113 and Line 118-119, LsSP1 band detected in rice plants infested by *L. striatellus* (Fig. 2C) seemed much stronger than that of in salivary sheath sample (Fig. 2D). It looks unreasonable as saliva secreted into rice plants should be much less than that of collected from artificial diet. Please clarify.

Response: Thank you for pointing out this. In the experiment to detect the secretion of LsSP1 into rice plants, we confined approximately 100 fifth instar nymphs in 2-cm stem, and the outer rice sheath with many salivary sheath attached were used for WB assay. Therefore, the LsSP1 band in this sample was very strong. Similar results were also found in previous reports. i.e., NIEG1 and NISPI secreted into plants by *N. lugens* (Ji et al., Plant Physiol, 2016; Huang et al., Front Plant Sci, 2020). For detecting the LsSP1 in artificial diet sample, although approximately 1000 nymphs were used, not all salivary sheath can be dissolved, as salivary sheath was very difficult to dissolve at room temperature. Also, planthopper might secrete salivary sheath of fewer branches in artificial diet than that of rice plant (Wang et al., 2008, 10.1111/j.1570-7458.2008.00785.x), which might result in few secretion of LsSP1 in artificial diet.

In the revised version, we detected LsSP1 in untreated plants, plants infested by *L. striatellus*, watery and gel saliva in the same gel, and similar results were found (Fig. 2C). Also, we described the procedure used in these two experiments in Line 543-558 and Line 566-569 in detail.

2. Line 153-154. Can the authors provide other LsSP1 candidate interactors revealed by Y2H screening?

Response: Done. In the revised version, we list seven LsSP1 candidate interactors in Table S2.

3. The interaction between LsSP1 and LsMLP is critical to the authors' finding. Please validate their interaction with more experimental methods such as Co-IP and luciferase complementation assay.

Response: Thank you for your suggestion. In the revised version, additional luciferase complementation assay was used to further validate the interaction between LsSP1 and LsMLP, and the result showed that LsSP1 can interact with LsMLP. We provide this results in Fig. 3G,H.

Also, we tried more methods to validate this interaction, including pull down and Co-IP. However, LsMLP cannot be expressed in *Escherichia coli*. For expression of recombinant LsMLP (fused with GFP or flag tags at N terminal or C terminal) in *N. benthamiana*, significant cell death was observed, similar with the reports of NIMLP (Shangguan et al., Plant Physiology, 2017), indicating that LsMLP can be expressed in *N. benthamiana*. However, the recombinant LsMLP cannot be detected by WB assays, even enriched by flag- or GFP-beads, respectively (Note: fluorescence signal can also be detected in the recombinant protein of LsMLP-GFP, but cannot be detected by WB). It is possible that MLPs cannot be abundantly accumulated in *N. benthamiana*, with its amount below the threshold for WB detection. It is worth to mention that in Shangguan et al., Plant Physiology (2017), the author also not perform WB assay for NIMLP detection, possibly that NIMLP cannot be detected by WB, either.

In this study, a total of 4 methods were used to validate the interaction between LsSP1 and LsMLP, including Y2H, BIFC, LUC, and NIMLP-knockdown. It might be reasonable to propose this interaction.

4. Can the authors provide Oryzain gene accession number in GenBank or locus identifier in Rice Genome Annotation Project Database? In Line 849-850, the authors indicated the OsOryzain identified by Y2H screening with arrow, but which can't be found in Figure S5. This could also be applied to LsSP1, although the authors provided nucleic acid and amino acid sequence of LsSP1 in Figure S1.

Response: Done. In the revised version, the OsOryzain gene accession number in GenBank and locus identifier in Rice Genome Annotation Project Database was provided in the manuscript. Also, the OsOryzain were highlighted by arrow in Figure S6. For LsSP1, the sequence has been submitted to GenBank, and the accession number (ON322955) has been provided in the manuscript.

5. As indicated by Fig 5 A and B, it looked like that *L. striatellus* infestation was more effective to induce the expression of OsOryzain and OsAALP-1 than that of SA treatment. Can the authors explain it in a reasonable way?

Response: Thank you for pointing out this. The transcript level of OsOryzain reached a peak at 6 h post *L. striatellus* infestation, while reached a peak at 12 h post SA treatment (Fig. 5A). The different induction patterns of two treatments indicated that SA treatment alone cannot totally be responsible for *L. striatellus*-induced OsOryzain expression. Other factors such as wound caused

by stylet penetration, interaction between saliva and other pathways might also be associated with *L. striatellus*-induced OsOryzain expression. In the revised version, we discussed this result in the discussion section in Line 380-383 as follow: “The transcript level of *OsOryzain* reached a peak at 6 h post *L. striatellus* infestation, while a peak was reached at 12 h post SA treatment (Fig. 5A). The different induction patterns indicated that other factors, in addition to SA pathway, might be also responsible for *L. striatellus*-induced *OsOryzain* expression, which deserved further investigation.”

6. In Line 745-747, the protein level of OsOryzain in rice sheath (C) in response to SA and *L. striatellus* (L.s.) were determined at 12 and 24 h post treatments (Fig 5C). Did the authors perform the WB assay of these protein samples on the same gel? If not, I am suggesting the authors compare the protein level of OsOryzain in the same gel (or blot). This could also be applied to Fig 5D.

Response: Thank you for your suggestion. In the revised version, OsOryzain in response to SA and *L. striatellus* were determined at 12 and 24 h post treatments on the same gel. Four replicates were performed for OsOryzain in rice sheath and rice apoplast, respectively, and similar results were found (Fig. 5B, C).

7. Line 273-275, the expression of NbPR1 was not significantly suppressed in the presence of LsSP1-mCherry in Fig 7E, please revise it.

Response: Done

8. Please indicate statistically significant differences in Fig 8A.

Response: Done

Reviewer #2 (Remarks to the Author):

The authors present a very thorough study into how two planthopper proteins interact with the rice immune system, leading to lowered cysteine protease-mediated plant defenses. The design of the study is elegant and the results are exciting. In my opinion, this study constitutes an important advance in the field. However, there are a few aspects of the manuscript that I think could be improved:

Response: Thank you very much for your comments on our manuscript. We have revised manuscript according to your recommendation.

Title and throughout the manuscript

Evolution has no purpose, traits do not evolve with a preconceived aim. The title and manuscript text need to undergo rigorous editing to reflect this better.

Response: Thank you for your suggestion. In the revised version, we removed the words “Fierce arm races” in the title. Also, the use of words “co-evolution” and “arm race” were carefully checked throughout the manuscript.

Abstract

Lines 23-25: The authors use the phrase “fierce arms races” here and elsewhere in the manuscript, but they do not present data on the evolutionary dynamics in this study system. For example, the authors only study a single genotype of the insect and a single genotype of the plant. The authors do not present evidence of arms race dynamics being at play in this system? Arms race dynamics can occur when co-evolution takes place in species interactions, but a lot more work would be required to check if co-evolution is taking place here. See for example the paragraph “Conditions for co-evolution” in the following paper: <https://www.nature.com/articles/ng1202-569>

Response: Thank you for pointing out this. Our study illustrated complex interactions between *Laodelphax striatellus* and *Oryza sativa*, and did not present evidence of arms race dynamics. According to the definition of “co-evolution” and “arm race”, it might not very suitable to use these phrases. Therefore, in the revised version, we carefully check the phrase “arms races” and “co-evolution” in the abstract and throughout the manuscript, and used the word “complex interaction” instead.

Introduction

Line 33: The nematodes that the authors refer to are herbivores as well? And which other herbivores do the authors refer to here?

Response: Thank you for pointing out this. In the revised version, we use the word “insects” instead. The sentence was re-written as “In nature, plants are continuously challenged by bacteria, fungi, nematodes, and insects” in Line 31.

Line 36: There is more than one non-self molecule.

Response: Thank you for pointing out this. PRRs can perceive multiple “non-self” molecules, and we re-wrote this sentence as “Pattern recognition receptors (PRRs) can perceive the “non-self” molecules known as pathogen-, herbivore- and damage-associated molecular patterns (PAMPs/HAMPs/DAMPs), and activate the pattern-triggered immunity (PTI), including mitogen-activated protein kinase (MAPK) cascades, reactive oxygen species (ROS), and hormone signaling” in Line 33-37.

Line 48-49: Line 48, These effector proteins were all characterized in the aphid *Myzus persicae*, so the authors might as well mention the species. The effectors do not negatively affect insect reproduction necessarily. This depends on the host plant-insect combination. It would be helpful if the authors could mention the host plant species and genotypes as well. The authors do not introduce what the Mp molecules are either, a reader needs more information for this paragraph to make sense. Line 49, Which plant species and genotypes can recognize these proteins? And again, the authors have not introduced that these molecules are proteins.

Response: Thank you for your suggestion. In the revised version, we addressed the species and corresponding plants when give examples that salivary protein might hinder insect performance by activating plant defenses. Also, we introduce that the molecule we mentioned are proteins. We rewrote these sentences in Line 44-49 as follow: “These oral secretions, on the one hand, hinder insect performance by activating plant defenses. For example, salivary protein Cathepsin B3 from *Myzus persicae* can be recognized by tobacco plants, which thus suppresses aphid feeding by

triggering ROS accumulation. Moreover, salivary protein 1 from *Nilaparvata lugens* induces cell death, H₂O₂ accumulation, defense-related gene expression, and callose deposition when it is transiently expressed in tobacco leaves or rice protoplasts”.

Line 62: Does the salivary sheath contain a single elicitor of plant defense or multiple elicitors?

Response: Salivary sheath is composite of multiple salivary sheath proteins. Currently, only the mucin-like protein is reported to be an elicitor. However, as functions of other proteins were unknown, there might have other components that also act as elicitors. Therefore, in the revised version, we re-wrote the sentence in Line 62-63 “Salivary sheath is composite of many salivary sheath proteins, which can be potentially recognized as HAMPs that activates the immune response in host plants”.

Lines 63-64: The authors need to be more precise. It has not been shown in the paper that the authors refer to that the salivary sheath evolves faster. That paper only suggests that a few proteins in the salivary sheath may evolve faster.

Response: Sorry for inadequate description. In the revised version, we re-wrote the sentence as “Because of their forward roles in herbivore–plant interactions, a few proteins in the salivary sheath may undergo a high evolution rate” in Line 64-65.

Results

Line 91-94: Can the authors provide more detail on how they identified LsSP1 as a candidate gene? Has the data that the authors refer to been published previously? Are the proteins specific to *L. striatellus* or do they also include other planthopper species such as the brown planthopper *Nilaparvata lugens* and the white-backed planthopper *Sogatella furcifera*?

Response: Thank you for your suggestion. In the revised version, we describe how to identify LsSP1 as a candidate gene. In detail, we describe the procedure in selecting the salivary gland-abundant, planthopper-specific, salivary gland-specific genes in Method section. A total of 30 genes met these criteria. We described these results in Result section and provided corresponding data in Table S1 and Fig. S1. Also, the corresponding sequences were provided in Source Data file. For selecting of LsSP1, we mentioned in the manuscript that this gene is among the top 5 most abundant genes. Also, LsSP1 or its homologous genes in other planthopper species have never been functionally investigated before.

In the revised version, we describe the selection procedure as follow: “As discovered when analyzing the transcripts highly expressed in *L. striatellus* salivary glands, many genes were planthopper-specific and their homologous genes were not found in other species (Table S1). To unveil their specific roles in the planthopper-rice interactions, this work firstly investigated the expression patterns of these genes in different tissues. A total of 30 genes were found to be specifically expressed in salivary glands (Fig.S1). The *L. striatellus* salivary protein 1 (hereafter: LsSP1, accession number: ON322955) was among the top 5 most abundant, salivary gland-specific, and planthopper-specific genes, which was thereby selected for further analysis” in Line 91-88. “Homologous analysis demonstrated that LsSP1 was a planthopper-specific protein, and exhibited 43.9% and 58.5% amino acid sequence identities to secretory proteins in the brown planthopper *Nilaparvata lugens* (ASL05017) and the white-backed planthopper *Sogatella furcifera* (ON322954), respectively (Fig. S2B). Noteworthy, LsSP1, as well as its homologous

genes in other planthopper species has never been functionally investigated before” in Line 110-113.

Lines 95-97: It would be helpful for the reader if the authors could provide information on the plants that the insects were reared on.

Response: Thank you for your suggestion. In the revised version, we addressed that we reared dsRNA-treated *L. striatellus* on ASD7 rice plants. Also, we clarified the rice genotypes used in each experiment in the Material Section and Figure legends.

Line 99: Perhaps use the word “significant” rather than the word “remarkable”?

Response: Done.

Lines 155-156: Could the authors provide more detail here? How was the screen performed? Were there proteins other than OsOryzain that were identified?

Response: Done. In the revised version, we describe the method used in screening the LsSP1 candidate interactors in Line 621-629 in detail, and list seven candidate interactors in Table S2.

Line 174: Should it be OsSAG12-1 instead of OsSAG-1? And should the start of the new sentence say “In addition to OsOryzain...” instead of “Besides...”? Is there an explanation why LsSP1 binds to some PLCPs but not others? It looks like the PLCPs it does not bind to do not form a monophyletic clade?

Response: Thank you for your suggestion. In the revision, more PLCPs were selected for investigating the interactions between PLCPs and LsSP1. In total, 15 PLCPs from 9 subfamilies were tested. We found that in addition to OsOryzain, another 9 PLCPs were confirmed to interact with LsSP1. However, when examined the relationship between PLCP phylogeny and their binding capacity to LsSP1, we did not find strict correlation. Also, there is no strict correlation between PLCP phylogeny and PLCP induction patterns. We presumed that the interaction between PLCP and LsSP1 is complex, which needs further investigation.

In the revised version, we summarize the results associated with PLCPs in Fig. S6, which include the phylogeny, the structure, the induction patterns, and the PLCP-LsSP1 interaction. Also, we re-described the results of PLCP-LsSP1 interaction in Line 180-191 as follow: “Altogether 46 canonical PLCPs were identified by searching the *O. sativa* genome, which were grouped into 9 subfamilies based on their homology to the previously categorized PLCPs (Fig. S6). Since PLCPs shared conserved peptidase domains (Fig. S6), we examined the potential interaction between LsSP1 and PLCPs from other subfamilies. In total, 15 PLCPs from 9 subfamilies were tested. In addition to OsOryzain, another 9 PLCPs were confirmed to interact with LsSP1, including Os04g57440, Os09g27030, Os05g24550, Os04g24600, Os05g01810, Os08g44270, Os09g39060, Os09g38920 and Os04g01710 (Fig. S6, S7). Noteworthily, there was no direct correlation of PLCP phylogeny with PLCP-LsSP1 interaction. For example, three PLCPs (Os08g44270, Os09g39170, and Os09g39060) from CEP1 subfamily showed different binding abilities to LsSP1. The binding specificity between PLCPs and LsSP1 deserves further investigation. Nonetheless, our experiments suggested that LsSP1 interacted with multiple PLCPs belonging to different subfamilies”.

Lines 180-190: Why did the authors decide to focus on OsAALP-1 and not the other PLCPs? Was expression of the other PLCPs tested as well and was expression of these also induced after planthopper feeding and/or SA treatment? Without this information it is impossible for a reader to judge if the expression pattern of OsOryzain is “typical” for a PLCP as mentioned in line 190.

Response: Thank you for pointing out this. In the revised version, we detect the expression patterns of 46 PLCPs to *L. striatellus* infestation and SA treatment using qPCR. There were 8 and 7 PLCPs found to be induced by *L. striatellus* infestation and/or SA treatment, respectively (Fig. S6 and Fig. S8). In this study, we selected OsOryzain for further study as **1) OsOryzain was fiercely induced by both *L. striatellus* infestation and SA treatment; 2) OsOryzain was firstly screened out in Y2H assay, and it strongly interact with LsSP1; 3) The availability and high specificity of OsOryzain antibody; 4) The homologous gene of OsOryzain in tomato (C14) was previously reported to mediate plant immunity.** Considering the selection criteria of PLCP, it might not very rigorous to use the word “typical”. Therefore, in the revised version, we summarize the induction patterns of PLCPs in Fig. S6, and addressed that “numerous PLCPs might be associated with the *L. striatellus*-induced, SA-mediated plant defenses in rice plants, and OsOryzain, with its homologous gene C14 in tomato reported to mediate plant immunity, was selected for further analysis” in Line 201-203.

Line 251 and others: It is “mCherry” and not “mCheery”?

Response: Done

Line 275: It appears that expression of NbPR1 is significantly suppressed in the presence of LsSP1-mCherry?

Response: Sorry for wrong description. The expression of NbPR1 was not significantly suppressed in the presence of LsSP1-mCherry. In the revised version, we corrected the description.

Lines 313-316: This description of the PCA appears too simplistic? oeSP1, and koOry are very similar to one another indeed, but they both are a bit different from WT in both treatment groups?

Response: Thank you for pointing out this. In the revised version, the description of the PCA was rewritten, and we emphasis the associated factors that explained the PC1 and PC2. Also, correlation analysis among 18 transcriptomic data was performed and illustrated in supplementary Fig. 13. These results were described in Line 324-331 as follow: “As revealed by principal component analysis (PCA), the first principal component (PC1) made the greatest contribution (62%) to variation, and PC1 could be explained as *L. striatellus* infestation. The second principal component (PC2) made 14% contribution, which showed that oeSP1 and koOry plants were closely associated with each other compared with that of WT plants, no matter whether they were untreated or ds*LsSP1*-infested (Fig. 9B). Similar results were found in correlation analysis, except for the fact that the expression patterns of *oeSP1* and *koOry* plants after ds*LsSP1*-infestation were not closely associated compared with that of WT plants (Fig. S13)”.

Discussion

Line 342: The fact that an effector protein suppresses defense in one host, but triggers it in another, does not necessarily indicate that the effector has opposite roles in both hosts, it could just have an

opposite effect.

Response: Sorry for inadequate description and the misleading. This paragraph of discussion mainly focused on the different roles of different salivary protein in one species in plant-insect interaction, and proposed that the function of saliva should be investigated from the perspective of multiple salivary components. In the revised version, we rewrote this sentence in Line as follow: “In previous studies, most salivary proteins are investigated individually, and it is found that different salivary proteins from one species have diversified functions. For example, in *M. persicae*, overexpression of salivary protein Mp10 activates multiple defense pathways in tobacco plants and reduces aphid performance; while overexpression of another salivary protein Mp55 increases the attraction of tobacco to aphids, and promotes aphid performance”.

Methods

Line 395: The authors use multiple rice genotypes for their experiments. Which genotypes were used in each experiment?

Response: Thank you for pointing out this. In the revised version, we clarified the rice genotypes used in each experiment as follow: “Two rice varieties (cv. ASD7 and Nipponbare) were used in this study. As the transgenic rice plants generated in this study were of Nipponbare background, we used wild type Nipponbare plants as a control. Therefore, the rice variety used in transgenic rice plant analysis (EPG and transcriptomic analysis in Fig. 9) was Nipponbare. For the rest of rice-associated experiments, ASD7 plants were used.” Also, the rice genotypes used in each experiment was described in figure legend.

Line 538: More detail on transcriptome sequencing and analysis if the transcriptome data needs to be provided.

Response: Done. In the revised version, the detail procedure used for transcriptome sequencing and analysis were provided. Also, the raw data generated in transcriptome sequencing and the TPM expression values generated in transcriptome analysis were submitted to NCBI SRA (PRJNA833487 and PRJNA815455) or provided in the Source Data file, respectively.

I encourage the authors to proofread the manuscript again for spelling and grammar.

Response: Done. In the revised version, we carefully checked the manuscript, and consulted an English Language Editor Service for modifying English writing again.

Reviewers' Comments:

Reviewer #1:

Remarks to the Author:

The revised manuscript clearly illustrated complex interactions between plants and herbivorous insects and will be a great contribution to the field.

I still have two concerns.

1. According to new information provided by the authors, the rice variety cv. ASD7 was used in this study, which appeared in Figs 1, 5, 6, 8, S8 and S9. As we know, ASD7 contains BPH (brown planthopper) resistance gene BPH2 and shows resistance to *Nilaparvata lugens*. Is ASD7 resistant to the small brown planthopper *Laodelphax striatellus*? Please clarify why ASD7 was used in this study.

2. As the authors claimed, *OsOryzain* plays positive role in plant immunity and *LsSP1* suppresses plant defenses. I am interested in how *oeSP1* or *koOry* plants react to *Laodelphax striatellus* infestation. Will they be more susceptible to *Laodelphax striatellus* than that of wild type plants?

I have no other comments.

Reviewer #2:

Remarks to the Author:

The authors have gone to great lengths to address the feedback from the reviewers and I am satisfied with nearly everything the authors have done to do that. I think this updated version of the manuscript is much improved and amounts to an impressive achievement. It is exciting that the authors have uncovered these molecular mechanisms underlying rice-planthopper interactions and they have done so expertly.

There are only several more minor, but nonetheless important, things left that I think deserve further attention.

Title

The word "to" in the title implies that evolution has a purpose, which it does not. Traits do not evolve with a preconceived aim. It would be more accurate if "to" would be changed to a word such as "that".

Abstract

The same applies to the word "to" in lines 13 and 23 of the Abstract.

Discussion

Lines 355-359: Coming back to my previous point on this. It is most likely not a "function" of salivary protein to induce defense. The induction of defense is more likely to be the consequence of the salivary protein being directly or indirectly recognized by the plant upon which a defense response is induced.

I am not sure which English Language Editor Service the authors have consulted, but many sentences still feel clunky to me and there are still many grammar mistakes left. This is doing the authors' elegant experimental work a disservice. Some mistakes are perhaps left in because the editorial service does not provide an editor with specific knowledge of this field of life sciences?

Reviewer #1:

The revised manuscript clearly illustrated complex interactions between plants and herbivorous insects and will be a great contribution to the field.

I still have two concerns.

1. According to new information provided by the authors, the rice variety cv. ASD7 was used in this study, which appeared in Figs 1, 5, 6, 8, S8 and S9. As we know, ASD7 contains BPH (brown planthopper) resistance gene BPH2 and shows resistance to *Nilaparvata lugens*. Is ASD7 resistant to the small brown planthopper *Laodelphax striatellus*? Please clarify why ASD7 was used in this study.

Response: Thank you for pointing out this. The resistant variety ASD7 was also reported to confer resistance to small brown planthopper, and was widely used for insect bioassays. Also, we also use the wild type Nipponbare plants to test the effect of ds*LsSP1* treatment on *Laodelphax striatellus* (Fig. 9), and very similar results were found. Therefore, in the revised version, we introduced that “The resistant variety ASD7, which contained the brown planthopper resistance gene BPH2, was also reported to confer resistance to small brown planthopper (Duan *et al.*, 2008; Duan *et al.*, 2009), was extensively applied for insect bioassays”.

2. As the authors claimed, OsOryzain plays positive role in plant immunity and LsSP1 suppresses plant defenses. I am interested in how *oeSP1* or *koOry* plants react to *Laodelphax striatellus* infestation. Will they be more susceptible to *Laodelphax striatellus* than that of wild type plants?

Response: In the revised version, we test the resistance of transgenic plants to *L. striatellus* infestation. The results showed that *oeSP1* and *koOry* plants showed a slight increase, but not statistically significant, susceptible to *L. striatellus* infestation (Fig. S14).

I have no other comments.

Response: Thank you very much for your suggestion.

Reviewer #2:

The authors have gone to great lengths to address the feedback from the reviewers and I am satisfied with nearly everything the authors have done to do that. I think this updated version of the manuscript is much improved and amounts to an impressive achievement. It is exciting that the authors have uncovered these molecular mechanisms underlying rice-planthopper interactions and they have done so expertly.

There are only several more minor, but nonetheless important, things left that I think deserve further attention.

Title

The word "to" in the title implies that evolution has a purpose, which it does not. Traits do not evolve with a preconceived aim. It would be more accurate if "to" would be changed to a word such as "that".

Response: Thank you for your suggestion. In the revised version, the title has been changed to “Planthopper salivary sheath protein LsSP1 inhibits salivary elicitor-induced plant defenses” as

suggested by Editor.

Abstract

The same applies to the word "to" in lines 13 and 23 of the Abstract.

Response: Thank you for your suggestion. In the revised version, we delete these inaccurate descriptions.

Discussion

Lines 355-359: Coming back to my previous point on this. It is most likely not a "function" of salivary protein to induce defense. The induction of defense is more likely to be the consequence of the salivary protein being directly or indirectly recognized by the plant upon which a defense response is induced.

Response: Thank you for pointing out this. In the revised version, we use "roles in insect-plant interactions" instead of "function", and rewrote this sentence as "different salivary proteins from one species have diversified roles in insect-plant interactions"

I am not sure which English Language Editor Service the authors have consulted, but many sentences still feel clunky to me and there are still many grammar mistakes left. This is doing the authors' elegant experimental work a disservice. Some mistakes are perhaps left in because the editorial service does not provide an editor with specific knowledge of this field of life sciences?

Response: Sorry about this. In the revised version, we consulted another Language Service (BioYee Magic Trans, Tangshan, China) to check our language. We hope our manuscript improved after modification.